# The role of C-O-H-F-Cl fluids in the making of Earth's continental roots

S. A. Gibson ✉, C. J. Jackson, J. C. Crosby & J. A. F. Day

The cratonic 'roots' of Earth's major continents extend to depths of over 160 km and have remained stable for more than 2.5 billion years due to buoyant, refractory harzburgites formed by Archean mantle melting. However, mantle harzburgites from some global cratons (e.g., Kaapvaal, Siberia, Slave, Rae and Tanzania) show unusual orthopyroxene and silica enrichment, alongside titanium depletion, which cannot be explained by simple melting processes. The origins of the orthopyroxene-rich harzburgites are debated and include high-pressure melting residues, with komatiite melt interaction, or subduction-related silicic melts and fluids. To further investigate this we analysed volatile ($H_2O$, F, Cl) contents in Kaapvaal craton peridotites. Orthopyroxene-rich harzburgites, including a diamond-bearing sample, show elevated volatile contents, suggesting infiltration by supercritical C-O-H fluids—rich in silica, fluorine and chlorine and depleted in Ti—fluxed from subducted oceanic lithosphere (carbonated pelites, eclogites and serpentinites). These findings highlight the role of C-O-H-F-Cl bearing fluids in shaping cratonic lithosphere and offer a new framework for understanding craton evolution, mantle metasomatism and diamond genesis in early Earth.

Over 43% of the surface area of Earth's major continents is composed of ancient cratons[1] (Fig. 1). These are regions where the lithosphere extends to depths >160 km and which have been tectonically stable for the last 2.5 Ga. The long-term resistance of these thick roots to tectonic activity[2] has been linked to the presence of low-density, refractory peridotites known as harzburgites, which constitute 35 to 60% of the cargo of mantle xenoliths transported to Earth's surface by kimberlites[3]. Cratonic harzburgites are depleted in 'magmaphile' elements (e.g., Fe, Ca and Al) and are believed to represent low-density residues from extensive adiabatic decompression melting (30–40%) of the convecting mantle during the Archaean[4,5]. As they cooled, these buoyant residues aggregated to form the overlying rigid continental (lithospheric) mantle. Subsequent re-fertilisation of the continental mantle by melts and fluids, derived from either the convecting mantle or down-going slabs[6,7], has modified the mineralogy or caused chemical (cryptic) changes[8].

The cratonic roots of the Kaapvaal, Siberia, Slave, Rae and Tanzania cratons (Fig. 1) are notably enriched in orthopyroxene (>25%) and garnet (up to ~10 modal %)[5,9–17]. Figure 2a highlights this relationship

for the Kaapvaal craton, where some of the most extreme enrichments in orthopyroxene are observed. These orthopyroxene-rich garnet harzburgites typically equilibrated at temperatures below 1150 °C, and are rich in $SiO_2$ (>45 wt.%) but depleted in $FeO_T$ (<7.0 wt.%) and Ti (<200 μg/g) (data from this work and refs. 14,16,18,19). They exhibit a wide variation in Mg# (Mg/Mg+Fe × 100), which is expressed in the forsterite content of olivines (91.6–93.9) and Mg# of orthopyroxenes (93.7–95.2)[9,10,16,18]. Notably, orthopyroxene-rich garnet harzburgites exhibit lower bulk $MgO/SiO_2$ ratios (Fig. 2b) and higher Mg# compared to lherzolites and other harzburgite varieties.

Despite long-standing knowledge that orthopyroxene-rich garnet harzburgites found in many global cratonic xenolith suites deviate from the low-pressure melting (<3 GPa) trend observed in modern oceanic mantle residues (Fig. 3a)[14,20], their origin remains a topic of considerable debate[5,15,16,18,19,21–26]. The combined high Mg# and low modal % of olivine in orthopyroxene-rich garnet harzburgites relative to other types of cratonic mantle peridotites was initially interpreted as evidence of high-pressure melting (4 to 8 GPa)[27] but this requires physically implausible amounts of fusion (40 to 70%; Fig. 3a), even at

Dept of Earth Sciences, University of Cambridge, Cambridge, UK. ✉e-mail: sally@esc.cam.ac.uk

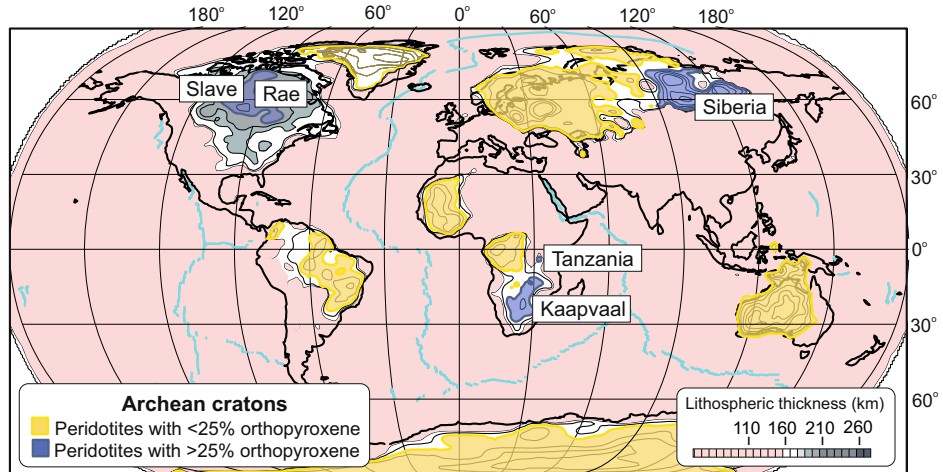

**Fig. 1 | Spatial distribution of orthopyroxene-rich garnet harzburgites in global Archaean cratons.** Not all cratons are made equal. Mantle peridotites with greater than 25% orthopyroxene cannot be explained as single-stage melt residues and require the addition of silica (data from this work, refs. 10,15,52,54,100). Contours for lithospheric thickness are shown at 10 km intervals for regions where this exceeds 160 km, and are taken from seismic tomography (modified from ref. 101, data from ref. 2). Outlines of shaded regions are selected on the basis of lithosphere >170 km so as to distinguish specific cratons from the more laterally extensive continental cores that are frequently composed of amalgamated cratonic blocks. For example, the Slave and Rae cratons amalgamated with the Superior and Wyoming cratons to form the North American craton and are located close to the thickest part of the continental core. The Kaapvaal craton forms the southern part of the larger Kalahari craton of southern Africa. Some of the deepest roots of Eurasia are associated with the Siberia craton. Mid-ocean ridges are illustrated by pale blue lines.

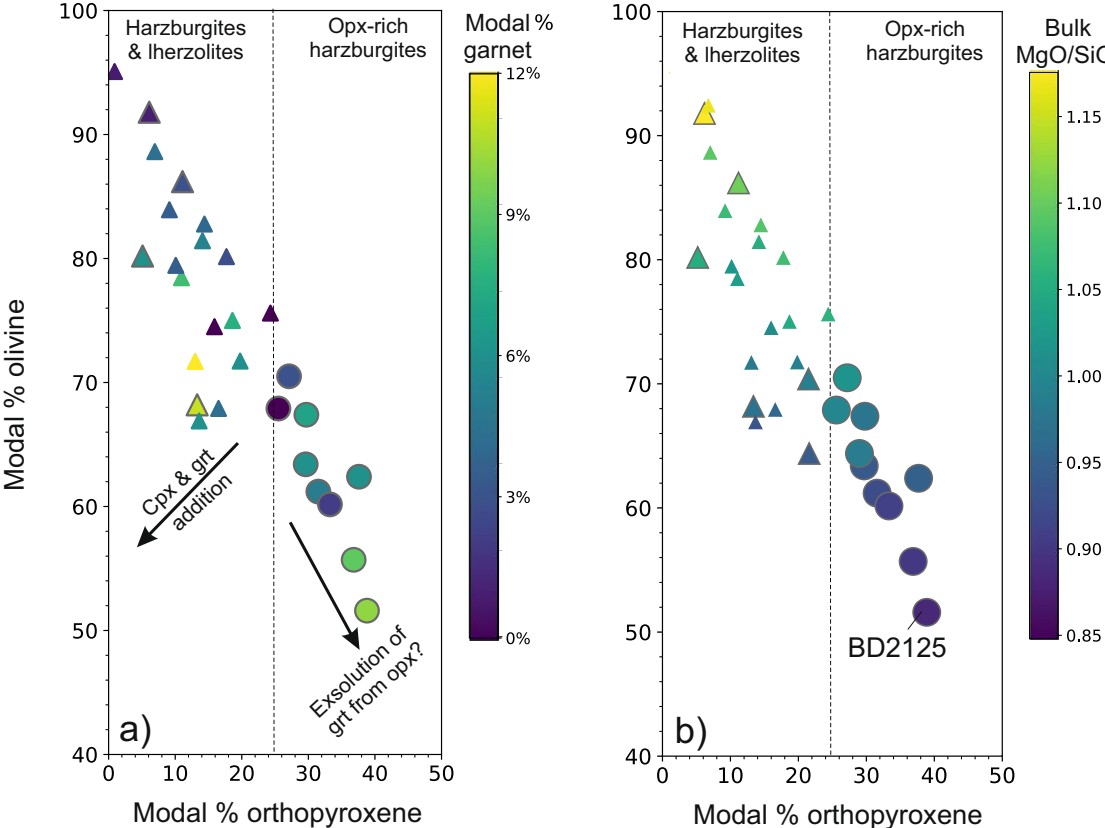

**Fig. 2 | Variation of modal amounts of olivine and orthopyroxene in the Kaapvaal craton.** Panel **a** shows that deviations from the main linear trend are primarily due to addition of garnet and or clinopyroxene to the melt residue as a consequence of metasomatism. The dashed line marks the threshold above which peridotites are described as having excess orthopyroxene compared to those in melt residues. Panel **b** shows that the modal amount of olivine in mantle peridotites primarily varies linearly with the amount of orthopyroxene and bulk MgO/SiO$_2$ content. The excess amount of orthopyroxene (>25%) primarily reflects the reaction of silica in a melt or fluid with olivine and can be described by the following equation: $MgFe_2SiO_4 + SiO_2 \rightarrow MgFe_2SiO_6$. Mantle peridotites from the Kaapvaal and Tanzania cratons studied in this work are shown by large symbols. Orthopyroxene-rich garnet harzburgites are shown by circles and garnet harzburgites and lherzolites are shown by triangles. Small symbols are for mantle peridotites from the Kaapvaal craton published by refs. 9,100.

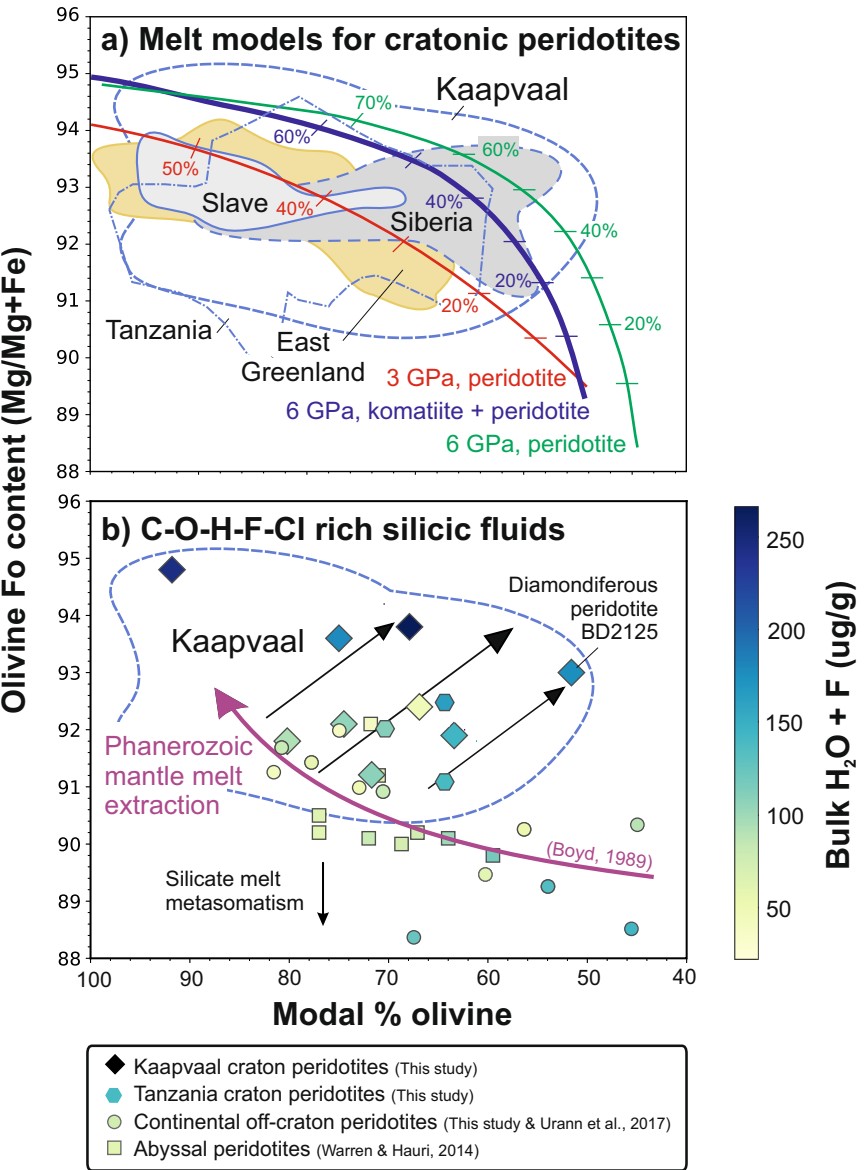

**Fig. 3 | Variation of modal abundance of olivine (volume %) with Fo content.**
The Fo contents (Mg/Mg+Fe) of olivines in mantle peridotites are thought to be the most reliable tracer of mantle melting. **a** Numerical models of mantle melting derived from experimental studies indicate that typical cratonic peridotites have experienced 20% or more melt extraction. Co-variations with the amount of modal olivine show that many of those from the cratonic mantle deviate away from the oceanic trend[14]. The Kaapvaal craton shows some of the greatest deviations and contrasts with the East Greenland craton[102], where there is a close approximation to the trend predicted for melts of peridotite at 3 GPa (red curve). While the deviation of cratonic mantle peridotites away from the oceanic and hypothetical 3 GPa melt residue trends may be explained by high-pressure melting (6 GPa, black curve), the amounts of fusion are implausibly high[22]. As a consequence, Tomlinson and Kamber[18] proposed that the anomalous enrichments in silica can be explained by the melting of a hybrid komatiite-peridotite source (blue curve), which requires lower amounts of melting. Isobaric melt models are after Tomlinson and Kamber[18]. Data are from this work (Supplementary Data 1), refs. 10,15,52,54,100. **b** Modal olivine versus Fo content plot together with the melt extraction trend for Phanerozoic mantle (purple arrow)[14] coloured for bulk $H_2O$ + F contents of cratonic mantle. Data for abyssal peridotites[54] and continental off-craton mantle (this work and ref. 52) are shown for comparison.

the high mantle potential temperatures ($T_P$) that existed in the Archaean[28]. A single-stage mode of formation for the orthopyroxene-rich harzburgites is also not supported by the presence of veins or segregations of orthopyroxene (Supplementary Fig. 1) and overgrowths of orthopyroxene on olivine[19] (Supplementary Fig. 2).

In addition to large enrichments in orthopyroxene, harzburgites from the Kaapvaal craton of southern Africa contain olivines with the highest Fo contents[5,9–16,18] (Figs. 1, 3). This cratonic mantle initially formed as residues from melting events between 2.5 and 3 billion years ago, which also coincides with the maximum age of formation of excess orthopyroxene (2.6 to 3.0 Ga)[16,19,23,29]. It has been proposed that the excess silica required to form orthopyroxene originated from reactions between fertile peridotite and melts of recycled eclogite[30], or repeated interaction of harzburgites in the continental mantle with komatiite melts formed at high pressures within hot upwelling mantle plumes[16,18,21,31]. These open-system melt-rock reaction processes require high-temperature thermal anomalies ($T_P$ = 1700–1800 °C) in the Archaean convecting mantle and contrast with theories that involve interactions between depleted continental mantle peridotite with subduction-derived silicic melts[5,9,22,25,32] or fluids[19,26], or hydrous within-plate silicic melts[23]. Another model suggests that the excess orthopyroxene formed through the hydration and serpentinisation of abyssal harzburgites prior to their subduction and incorporation into cratonic roots[33,34].

The diverse prevailing hypotheses for the formation of the excess orthopyroxene have important implications for our fundamental understanding of the tectonic processes involved in stabilising Earth's continental cores. While the various models that have been proposed appear to satisfactorily explain the excess orthopyroxene and bulk major element characteristics (e.g., low MgO/SiO$_2$)[5,18,19,21,24,26,30,31,33,34] of orthopyroxene-rich garnet harzburgites, existing datasets cannot readily be used to discriminate between them. Hence the ongoing controversy. The motivation behind our work is that each tectonic scenario that has been proposed for the origin of the excess orthopyroxene will imprint unique trace and volatile element (including C, H, F, Cl and S) signatures on the mantle. This is because of the variable compatibility of these elements during mantle melting and also different partitioning into melts and fluids. For example, large amounts of adiabatic decompression melting in the convecting mantle (~40%) during the Archaean would leave residues devoid of strongly incompatible elements (including C, H and F[35]) and more depleted than the present-day upper mantle MORB source[36] (see Supplementary Information), whereas multiple phases of enrichment would generate elevated, but variable, trace and volatile element concentrations[37,38]. Furthermore, peridotite and eclogite melts might be expected to have different volatile and trace element signatures to metasomatic melts or fluids fluxed from sediments, altered oceanic crust and serpentinites in down-going slabs of oceanic lithosphere.

Some orthopyroxene-rich garnet harzburgites are diamondiferous[39–41], which is important because lithospheric mantle diamonds are widely believed to form from metasomatic fluids[42,43], and it has been proposed that the formation of the excess orthopyroxene (>25%) and introduction of carbon are linked[23]. Here, we offer a new perspective on the formation of excess orthopyroxene and diamond formation in the roots of cratons by presenting high-precision in situ secondary-ion mass spectrometry (SIMS) analyses of multiple volatiles (H$_2$O, F and Cl), together with major and trace elements of the most abundant phases in mantle peridotites predominantly from the Kaapvaal and also Tanzanian cratons (Supplementary Fig. 3), where some of the most extreme enrichments of orthopyroxene and silica occur (Fig. 3a)[5,12,14,15,18].

## Results

### Major and trace element chemistry

The bulk compositions of Kaapvaal orthopyroxene-rich garnet harzburgites in our study are relatively silica-rich, with MgO/SiO$_2$ ratios between 0.89 and 0.94, compared to the global range of 0.85 to 1.06[18]. Kaapvaal sample BD2125 is notably more magnesian than the other orthopyroxene-rich garnet harzburgites, with olivine at Fo$_{93}$ and orthopyroxene Mg# at 93.9 (Supplementary Data 1, Supplementary Fig. 4) but, despite this, has the lowest bulk MgO/SiO$_2$ ratio (0.89) in our entire analysed xenolith suite (Supplementary Data 2 and Fig. 2a). This is primarily because of its high orthopyroxene content. Additionally, phlogopite in this orthopyroxene-rich garnet harzburgite has higher Cr$_2$O$_3$ (0.82 wt.%) but lower TiO$_2$ contents (0.12 wt.%) than found in other lithologies Kaapvaal lithologies (e.g., this work, ref. 44) (Supplementary Data 4).

The garnets in Kaapvaal orthopyroxene-rich harzburgites are subcalcic and differ from those found in other mantle lithologies by their low contents of Ti and Y, together with sinusoidal rare-earth element (REE) patterns that peak in the middle REEs (MREEs; Supplementary Fig. 7)[10,45]. Both the garnets and clinopyroxenes in these harzburgites are distinguished from those in other lithologies by their positive heavy-REE (HREE) slopes, i.e., between Ho and Lu[10]. Since garnet and clinopyroxene are key hosts of incompatible trace elements in the mantle, they influence the bulk compositions of these and other peridotite xenoliths (Fig. 4). On MORB-source[36] normalised multi-element plots, the reconstructed bulk compositions of orthopyroxene-rich garnet harzburgites exhibit depletions at high-

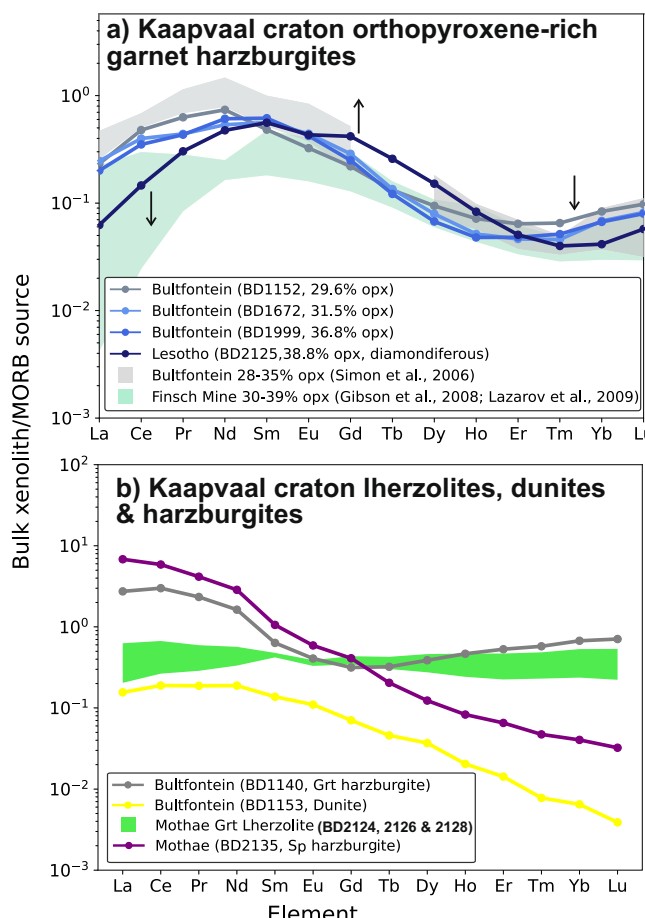

**Fig. 4 | Bulk rare-earth element (REE) concentrations of cratonic peridotites.** **a** Orthopyroxene-rich garnet harzburgites and **b** other types of peridotites in mantle xenolith suites from the Kaapvaal craton of southern Africa. Elements increase in incompatibility during mantle melting from right to left. The orthopyroxene-rich garnet harzburgites exhibit sinusoidal REE patterns when normalised to the MORB source mantle. These patterns vary with the amounts of orthopyroxene. As the amount of orthopyroxene increases, the contents of light and heavy REEs decrease and middle REEs increase (as illustrated by the black arrows). The most extreme example of this in our sample suite is the diamondiferous orthopyroxene-rich harzburgite BD2125 from Lesotho. This xenolith equilibrated just below the graphite-diamond boundary (1060 °C and 4.4 GPa). Data are from this work (Supplementary Data 1) unless stated otherwise in the legend. A MORB-source normalisation[36] is used because cratonic mantle is widely believed to represent aggregated residues of large amounts of melting.

field strength elements (i.e., Nb, Ta and Ti) together with Th and U but have high Pb concentrations (Fig. 5). In comparison to other lithologies in the cratonic mantle, the orthopyroxene-rich garnet harzburgites are depleted in Ti and Sr together with the light rare-earth elements (LREEs) and HREEs. They also have the lowest Ce/Pb ratios, which are less than 1 (0.03–0.73) times the MORB source. The diamondiferous orthopyroxene-rich garnet harzburgite (BD2125) exhibits the greatest bulk-rock depletion in LREEs and Sr, yet it has the highest contents of MREEs and Pb (Figs. 4, 5).

### Volatile element (H, F and Cl) chemistry

The volatile contents of the most abundant mineral phases (olivine and pyroxenes) in peridotites from the Kaapvaal and Tanzania cratons show considerable variability (Supplementary Fig. 5 and Supplementary Data 1). We use a broad definition of water that includes H incorporated into the crystal lattice of silicate minerals, reporting concentrations as μg/g H$_2$O[46,47]. Mantle garnets have relatively low

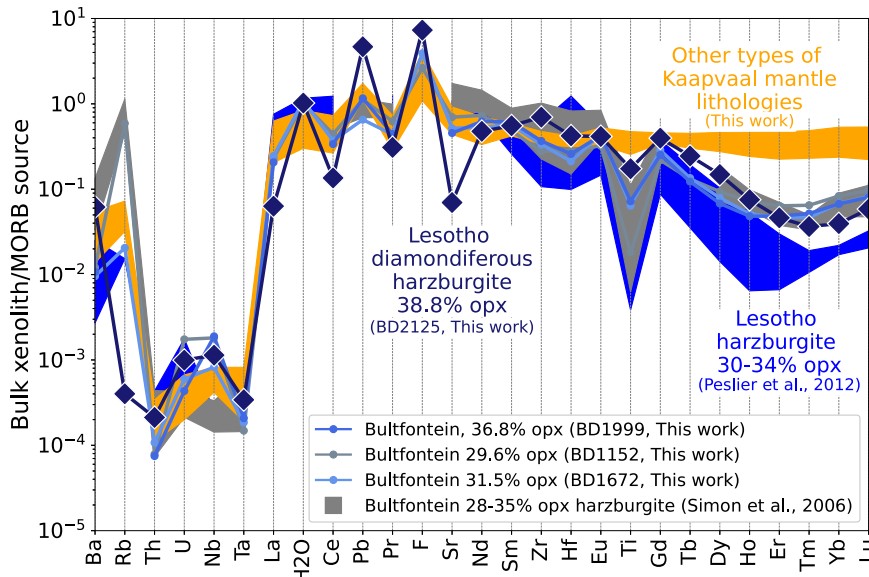

**Fig. 5 | Bulk xenolith concentrations of incompatible trace and volatile elements ($H_2O$ and F) in orthopyroxene-rich garnet harzburgites from the Kaapvaal craton of southern Africa normalised to MORB source upper mantle[36].** Elements increase in incompatibility during mantle melting from right to left. Incompatible trace element concentrations in our samples of orthopyroxene-rich garnet harzburgites resemble those in the dataset of ref. 9. All of the orthopyroxene-rich garnet harzburgites are depleted in Ti and the heavy rare-earth elements relative to other types of peridotites (lherzolites, harzburgites and dunites). Diamondiferous orthopyroxene-rich garnet harzburgite (BD2125) exhibits the greatest relative enrichments in $H_2O$, F, Pb, Th and U and depletion in Rb. Data are from this work (Supplementary Data 1) and ref. 9.

contents of water (mean = 11 μg/g[48]) and so were not analysed here. Statistical analysis of our new and published data from the Kaapvaal craton[10,48–50] (Supplementary Data 5) shows that olivines in the orthopyroxene-rich garnet harzburgites have similar contents of $H_2O$ (40 ± 26 μg/g) to those in garnet-bearing lherzolites (47 ± 49 μg/g) and garnet harzburgites (52 ± 43 μg/g) but lower than those in the dunite (81 μg/g) and much higher than olivine in the spinel harzburgites (7 ± 2 μg/g). The orthopyroxenes in the orthopyroxene-rich garnet harzburgites have similar high contents of $H_2O$ (174 ± 70 μg/g) to the garnet harzburgites (180 ± 73 μg/g) and dunite (203 μg/g) but are more hydrated than those in the garnet lherzolites (158 ± 72 μg/g) and spinel harzburgites (118 ± 97 μg/g). Clinopyroxenes in the orthopyroxene-rich garnet harzburgites also have higher $H_2O$ contents (280 ± 73 μg/g) than those in the garnet lherzolites (206 ± 97 μg/g), dunite (195 μg/g) and spinel harzburgite (74 ±5ug/g). Notably, the diamondiferous orthopyroxene-rich garnet harzburgite (BD2125) has high $H_2O$ in olivine (92 μg/g) compared to other orthopyroxene-rich garnet harzburgites (Supplementary Fig. 6). Of the most abundant phases in the cratonic mantle, olivine is the main host of fluorine[10]. In the orthopyroxene-rich garnet harzburgites, mean contents of F in olivine are lower (65 ± 50 μg/g) than the dunite (164 μg/g) but similar to the garnet lherzolites (47 ± 54 μg/g), garnet harzburgites (57 ± 40 μg/g), and spinel harzburgites (54 μg/g) (Supplementary Fig. 56 and Supplementary Data 5). Notable is the exceptionally high F content of olivine (135 μg/g) in the diamondiferous orthopyroxene-rich garnet harzburgite (BD2125; Supplementary Figs. 5, 6).

Because of the strong partitioning of F into olivine, the mean contents of F in orthopyroxene are low and similar in the orthopyroxene-rich garnet harzburgites (17 ± 7 μg/g), garnet lherzolites (20 ± 5 μg/g), garnet harzburgites (22 ± 10 μg/g), dunite (26 μg/g) and spinel harzburgites (9 ± 6 μg/g; Supplementary Fig. 6). The F contents of clinopyroxene in the orthopyroxene-rich garnet harzburgites (23 ± 8 μg/g) are intermediate between those of the garnet harzburgites (29 ± 7 μg/g), garnet lherzolites (38 ± 14 μg/g), dunite (46 μg/g) and spinel harzburgite (20 ± 6 μg/g). The most F enriched pyroxenes in the orthopyroxene-rich garnet harzburgites are found in BD2125 (Supplementary Fig. 6).

Cl contents are low in olivine and pyroxenes from all peridotites in the Kaapvaal and Tanzania cratons and most of our analyses are close to background levels (Supplementary Data 3). We note, however, that phlogopite in diamondiferous orthopyroxene-rich garnet harzburgite BD2125 contains higher Cl (0.17 μg/g; Supplementary Data 4) than found in other lithologies Kaapvaal mantle lithologies (<0.05 μg/g) (e.g., this work, ref. 44).

## Bulk volatile contents

The partitioning of volatiles (such as H, F and Cl) into the crystal lattice of mantle phases is complex and controlled by a wide variety of factors, including mineral chemistry, temperature, pressure, oxygen fugacity, melt or fluid composition and bulk assemblage[10,46,47,49,51–53]. Over the course of geological time, these variables will influence sub-solidus re-equilibration between co-existing phases and−given the age of samples in our study−it is crucial to examine bulk-rock as well as mineral volatile contents. Since olivine is typically the most abundant phase in mantle peridotites, its volatile content generally influences the overall bulk composition.

Statistical analysis of both our new and published reconstructed bulk volatile data for the Kaapvaal craton (Supplementary Data 6) reveals that the mean bulk $H_2O$ content of the orthopyroxene-rich harzburgites (83 ± 38 μg/g) is similar to the garnet lherzolites (71 ± 32 μg/g) and dunite (90 μg/g) but higher than the garnet harzburgites (59 ± 29) and spinel harzburgites (37 ±25ug/g; Fig. 6). Since the orthopyroxene-rich harzburgites have low Ce this gives rise to the highest bulk $H_2O/Ce$ (Fig. 5). The diamondiferous sample (BD2125) has exceptionally high bulk $H_2O$ (119 μg/g) and $H_2O/Ce$.

Akin to $H_2O$, the mean bulk F contents of the orthopyroxene-rich garnet harzburgites (51 ± 26 μg/g) are similar to the garnet lherzolites (36 ± 44 μg/g) and dunite (153 μg/g) but higher than the garnet harzburgites (48 ± 30 μg/g) and spinel harzburgites (23 μg/g; Fig. 6). Within the orthopyroxene-rich garnet harzburgites there is a positive correlation between bulk F content, modal % of orthopyroxene (Fig. 7) and bulk contents of $SiO_2$ and $Cr_2O_3$ but negative correlations with NiO and MgO. Moreover, bulk F contents increase as bulk LREE abundances decrease and the peak in the normalised REE pattern migrates from Nd

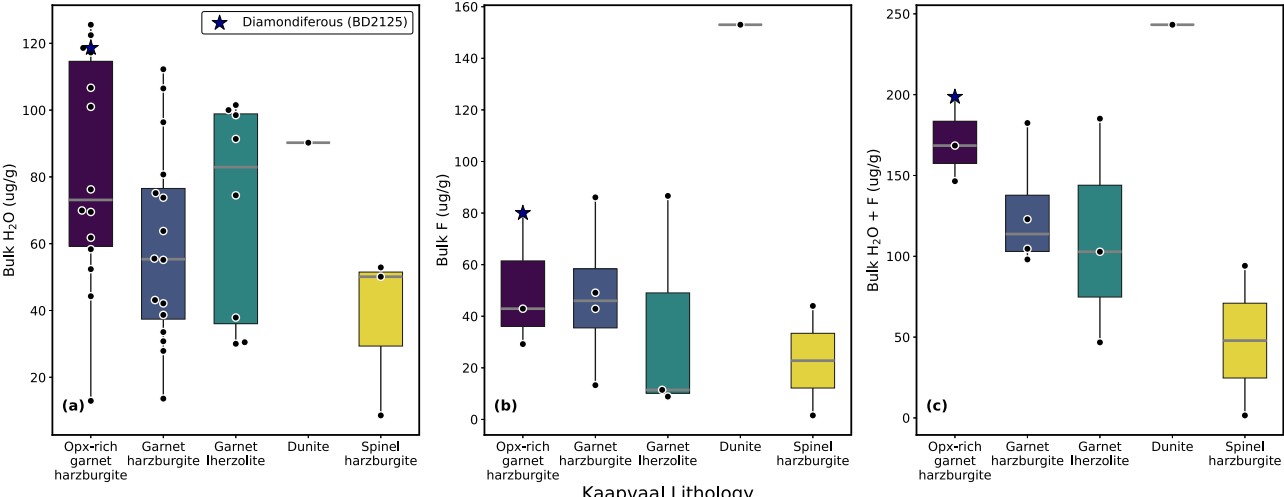

**Fig. 6 | Box and whisker plots showing variations in reconstructed bulk contents of volatiles in different types of mantle peridotites from the Kaapvaal craton.** Panel **a** shows the variations in $H_2O$, **b** F and **c** $H_2O + F$. The boxes define the edge of the 1st and 3rd quartile ranges, the grey line inside the box is the median value and the whiskers extend to 1.5× the interquartile range and represent the range of the data. Data are from this work and Peslier et al.[49]. Statistical analyses are available in Supplementary Data.

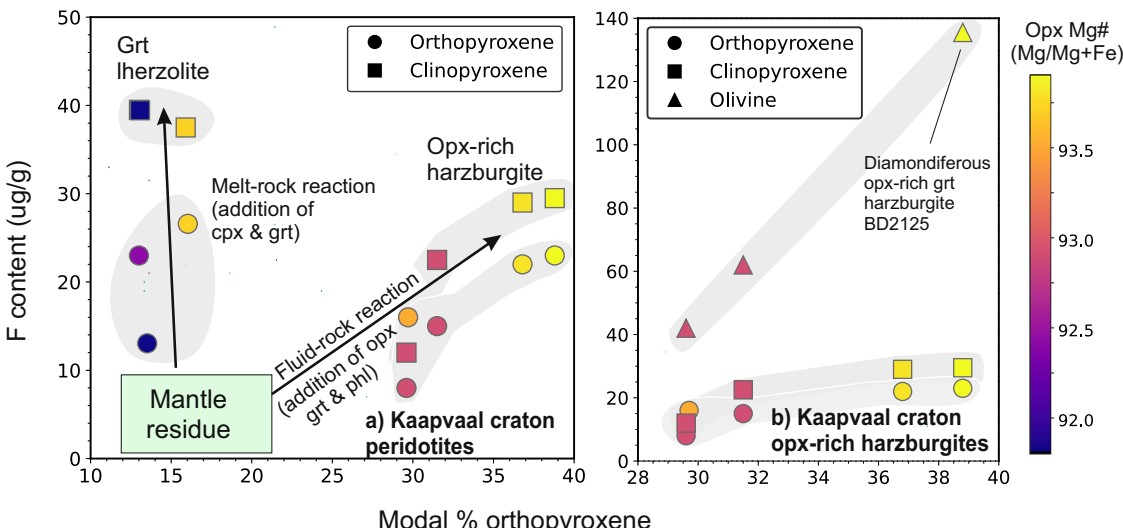

**Fig. 7 | Modal abundance of orthopyroxene (volume %) versus mean F content of mineral phases in Kaapvaal peridotites. a** F content of orthopyroxene and clinopyroxene in garnet lherzolites and orthopyroxene-rich harzburgites from the Kaapvaal craton. Vectors show the effects of melt- and fluid-rock reactions on mantle residues, which are depleted in F (e.g., depleted MORB mantle has 12 µg/g F)[103]. The volume of fluid controls the magnitude of the increase in excess modal orthopyroxene (garnet and phlogopite) and fluorine concentration. **b** Same as for (**a**) except the maximum value of the y-axis scale is increased to also include olivine, and the minimum value of the x-axis scale is increased so that only orthopyroxene-rich garnet harzburgites are shown. Data are from this work (Supplementary Data 1).

to Sm, reaching the most extreme in BD2125 (Fig. 4). This diamondiferous orthopyroxene-rich garnet harzburgite contains exceptionally high bulk F (70 µg/g; Fig. 6).

## Discussion
### Mantle melting models
Phanerozoic oceanic melt residues, including abyssal peridotites[54], show a decrease in bulk $H_2O + F$ with increasing forsterite content (Fo = Mg/(Mg+Fe) × 100) and modal abundance of olivine (Fig. 3b). This is consistent with the incompatible nature of both $H_2O$ and F during melting of peridotite. Continental off-craton peridotites show a similar trend to oceanic mantle for Fo content and bulk $H_2O + F$, which decreases from 175 to 50 µg/g (this work and ref. 52). As discussed above, the co-variations in Fo contents and amounts of modal olivine in some cratonic peridotites are offset from the low-pressure (<3 GPa)

oceanic melt residue trend. Our new data shows that as the magnitude of this deviation increases, the total bulk $H_2O + F$ budget also increases (to 270 µg/g). Samples that plot close to the high-pressure (6 GPa) melting trend[18] have the highest bulk $H_2O + F$ contents (Fig. 3b), including the diamondiferous orthopyroxene-rich garnet harzburgite (BD2125). A further observation from our dataset is that cratonic peridotites show a wide range in bulk $H_2O/F$ ratios (0.6 to 5), with BD2125 having a relatively low ratio ($H_2O/F = 1.2$) compared to other orthopyroxene-rich garnet harzburgites (up to 4). The Mg-rich dunite (BD1153) has the lowest $H_2O/F$ (0.6) in our sample suite.

Reconciling the high concentrations of Si, Mg, C, F and $H_2O$ plus the low Ti and Fe observed in orthopyroxene-rich garnet harzburgites with models involving high-pressure "auto-refertilisation" of fertile lherzolite by eclogitic melts[30] poses a challenge. Although this process might account for excess silica, both eclogitic melts and fertile

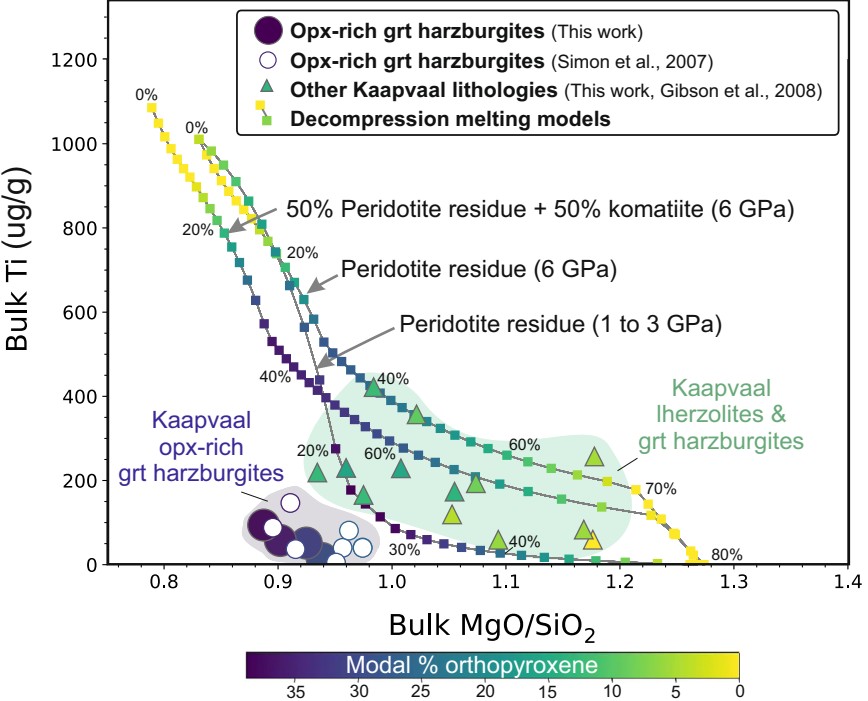

**Fig. 8 | Bulk MgO/SiO₂ versus Ti in cratonic mantle peridotites and melting residues predicted from thermodynamic models.** Curved lines show models for isobaric melting of mantle peridotite (KR4003) at 3 and 6 GPa; and reaction of komatiite melts with peridotite (KR4003) at 6 GPa[18]. The 2 and 3 GPa melting trends are similar[18]. The amount of melting associated with the residues is given as % values. All symbols are coloured according to the modal % of orthopyroxene. Only melting at 6 GPa generates the high Mg# of olivine and orthopyroxene observed in the orthopyroxene-rich harzburgites, but the Kaapvaal orthopyroxene-rich harzburgites have much lower bulk Ti contents than the residues predicted from 6 GPa melt models. An alternative mechanism that causes depletion in Ti and enrichment in Si is therefore required. Bulk MgO/SiO₂ data for peridotite xenoliths are reconstructed from EPMA analyses of minerals based on this work (Supplementary Data 1); Gibson et al.[5] and whole-rock XRF analyses (Simon et al.[9]). Bulk xenolith Ti (μg/g) contents are also reconstructed, but from LA-ICP-MS analyses, which are at a higher precision than those from EPMA. The highest MgO/SiO₂ (1.18) in our sample suite from the Kaapvaal craton occurs in a metasomatised dunite (BD2153).

lherzolites are typically enriched in Ti and Fe, contrasting sharply with the depleted Ti and Fe levels in these harzburgites. To investigate alternative potential sources of both volatile and non-volatile elements in the orthopyroxene-rich harzburgites, we employed rare-earth element (REE) inversion melting models[55]. These models use the REE contents of the peridotites to quantify the amount of mantle melting and then apply the same models to other trace elements, including volatiles[35]. The REE inversion models indicate that the pronounced positive heavy REE (HREE) slope, i.e., between Tm and Lu, observed in the orthopyroxene-rich garnet harzburgites can be reproduced by ~40% polybaric fractional melting of upwelling peridotite (Supplementary Fig. 8). While such high degrees of melt extraction would 'strip' the residue of elements more incompatible than the HREEs (including C, H and F), they are consistent with the low MgO/SiO₂ ratios predicted by thermodynamic models for orthopyroxene-rich harzburgite formation[18] (e.g., Figs. 3, 8). The extremely low concentrations of incompatible trace elements and volatiles that remain in the residue after such large melting events will, however, be readily masked by any subsequent infiltrating metasomatic melts and fluids.

### Reactive infiltration of Al-enriched komatiites

Recent studies propose that the formation of excess orthopyroxene in cratonic roots could result from the interaction of Al-enriched komatiite melts with ultra-depleted harzburgites[18,21]. Examples of such Al-enriched komatiites include Archaean occurrences from Commondale and the Buffalo River in South Africa[56,57]. These komatiites exhibit low Ti contents (~300 μg/g), high Al₂O₃/TiO₂ ratios (~80), elevated Pb concentrations (0.2–0.8 ppm), and are depleted in light and middle REEs relative to other komatiite types (Fig. 9a)[56–58]. It has variably been suggested that Al-enriched komatiites may result from local open-

system re-melting of moderately depleted lithospheric mantle at 5 GPa[21] or by ultra-deep melting (initiated at depths >600 km) in upwelling mantle plumes with a $T_P$ of 1850 °C[59]. While a 50:50 mix of depleted melt residues and Al-enriched komatiites can approximate the major element compositions of orthopyroxene-rich harzburgites[18], our trace element models reveal notable misfits for Ba, Rb, Ti and the HREEs (Fig. 9a).

Although no volatile data exist for the Al-enriched komatiites, other types of komatiites, spanning Archaean to Phanerozoic in age (e.g., Abitibi, Belingwe, Weltevreden and Gorgona), are known for their relatively high H₂O contents (~0.6 wt.%), elevated H₂O/Ce ratios and significant Cl concentrations (0.08 wt.%)[60–62] (Fig. 9a). The origin of this water (i.e., deep recycled vs surface source) is, however, highly contentious[58,61]. Moreover, unlike orthopyroxene-rich harzburgites, komatiites lack both the significant fluorine enrichment on normalised multi-element plots (Fig. 9a)[60,62] and the high concentrations of C required to form diamond; primary Gorgona komatiites, for example, are estimated to have 0.16 wt.% CO₂[63]. Below, we explore alternative sources for C and other volatiles in the orthopyroxene-rich garnet harzburgites.

### Infiltration of subduction-related melts and fluids

Silicic and carbonated aqueous and supercritical fluids as well as melts play a key role in the mass transfer of elements at subduction zones[64–68]. In the cratonic mantle, these subduction-related metasomatic agents are preserved in xenolith compositions and also as high-density fluids (HDFs) encapsulated in diamond[69]. The diverse range of compositions of these metasomatic melts and fluids is controlled by the protolith together with temperature, pressure and redox conditions in the down-going slab.

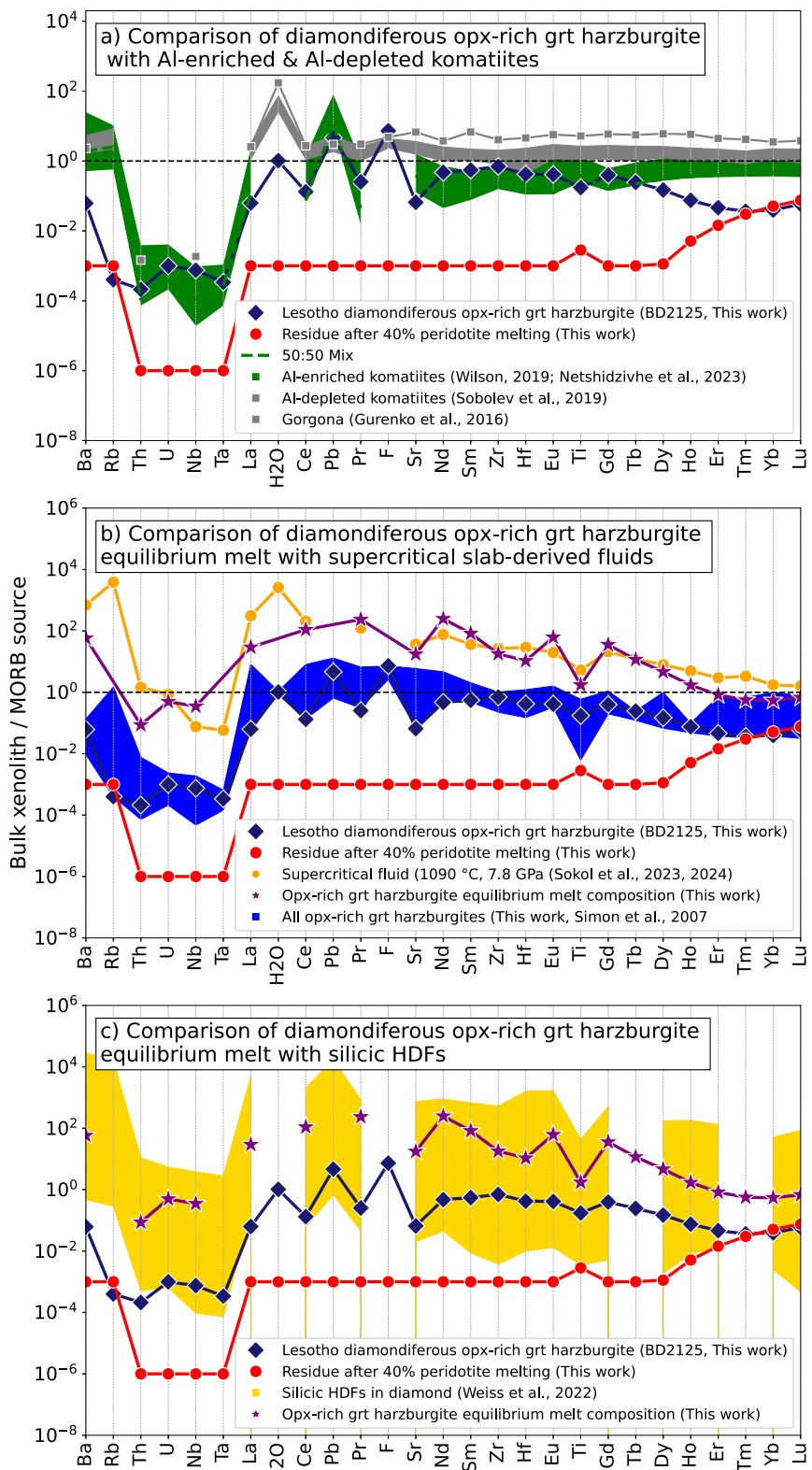

**Fig. 9 | Comparison of MORB-source normalised[36] incompatible trace and volatile element patterns in orthopyroxene-rich garnet peridotites from the Kaapvaal craton (this work and ref. 9) with potential sources of excess silica.** The compositions of the orthopyroxene-rich garnet harzburgites are compared with: **a** mean composition of Al-depleted and Al-enriched komatiites[61,62,104]; **b** subduction-related supercritical fluids generated in high-pressure experiments[66]; and **c** fluids with a similar composition to those found in diamond inclusions. The calculated concentrations of a 50%:50% mix of depleted melt residue and Al-

depleted komatiite are shown as a green dashed line on (**a**). (**b**) and (**c**) also show the calculated composition of a metasomatic agent in equilibrium with garnet[79] in orthopyroxene-rich garnet harzburgite BD2125. This assumes that 30% of the incompatible trace element content of BD2125 garnets is derived from the meta-somatic agent, consistent with mass balance calculations of the amount of excess silica (see text for discussion). Elements increase in incompatibility during mantle melting from right to left. Data are from ref. 10 unless stated otherwise in the legend.

Experimental studies related to subduction zones reveal that the breakdown of chlorite releases aqueous fluids at <850 °C and <5.5 GPa[70,71]. At pressures greater than 4.5 GPa and temperatures between 800 to 1100 °C, carbonated pelites melt to form C-O-H supercritical fluids, while higher temperatures and pressures lead to the formation of carbonated-silicate melts[72]. Hydrous and carbonated eclogites release aqueous fluids at slightly higher pressures and temperatures (4 GPa and 1000 °C), and these become supercritical at ~5.5 GPa and 1050 °C[68]. Recent experiments have shown that in a hydrous environment, the interaction of eclogitic C-O-H supercritical fluids with harzburgite increases the amount of orthopyroxene at the expense of olivine[68].

Notably, supercritical fluids and melts derived from carbonated pelites are capable of transporting substantial amounts of silica, as well as Mg, large ion lithophile elements (e.g., Ba, Rb, Th, U, Pb, and Sr), light rare-earth elements and volatiles such as $H_2O$ and $CO_2$ but, importantly, Ti remains immobile[64–68,73,74]. Subduction-related supercritical fluids exhibit similar partitioning behaviour to sediment melts but have higher Si/Al, $H_2O$ (~30 wt.%), La/Ta and lower Th/La and Sr concentrations[66,75,76]. While the trace element ratios of slab-derived supercritical fluids and sediment melts are subtly different, there are important variations in their viscosity. The lower viscosity of supercritical fluids in the mantle[77], combined with melt-like wetting properties and high element-carrying capacity, makes them ideal agents for chemical transport in cratonic mantle above subduction zones.

The extent to which subducted lithologies contribute to the different types of HDFs encapsulated in diamonds—namely saline, silicic, and carbonatitic—remains uncertain. Despite this, some HDFs can contain up to 65 wt.% $SiO_2$ and display a broad range of trace element concentrations. These vary from compositions similar to orthopyroxene-rich garnet harzburgite to those resembling supercritical fluids derived from subducted carbonated pelites (Fig. 9). Similar to slab-derived fluids, silicic HDFs enclosed in diamonds are relatively enriched in Ba and LREEs and depleted in Th, U, Nb, Ta, Sr, and Ti. Nonetheless, they exhibit lower $H_2O$ (9.8 wt.%) and $CO_2$ (4.2 wt.%) contents compared to supercritical fluids from subducted carbonated pelites, which typically contain ~30 wt.% $H_2O$ and ~10 wt.% $CO_2$[67].

If we assume the original concentration of $SiO_2$ in the depleted cratonic mantle is ~40 wt.% (Supplementary Data 1) and ~65 wt.% in the metasomatic fluid[67], mass balance calculations require the addition of approximately 30% of the silicic component to account for the excess $SiO_2$ in the orthopyroxene-rich garnet harzburgites (48 wt.%). The high silica content of the metasomatic agent would not be in equilibrium with peridotite and confirms an alternative protolith, such as eclogite or pelagic sediments[64–67,78].

We used published partition coefficients (D values) to estimate the composition of melts or fluids in equilibrium with garnet in the orthopyroxene-rich garnet harzburgites. While this approach does not consider other phases present in orthopyroxene-rich garnet harzburgites, it remains a reasonable approximation given that garnet is the primary host of trace elements in these clinopyroxene 'poor' peridotites. Currently, no D values exist for supercritical fluids derived from carbonated pelagic sediments or for high-pressure silicic fluids, so we applied the partition coefficients of Brey et al.[79] (Fig. 9) as a first-order approximation, with the caveat that these D values are based on experiments with carbonated melts and lack volatile data. We used the calculated equilibrium melt trace element composition together with that of a refractory mantle (depleted by 40% melt extraction) and supercritical and high-density fluids in mass balance calculations to obtain the best-fit for the observed concentrations in the orthopyroxene-rich garnet harzburgites (Fig. 9).

Despite the limitations, our trace element modelling suggests that the metasomatic agent in equilibrium with garnet in the diamondiferous orthopyroxene-rich garnet harzburgite (BD2125) could result from the interaction of a depleted peridotite residue with approximately 30% supercritical fluids derived from carbonated pelites or silicic high-density fluids (HDFs; Fig. 9b, c). Carbonated melts and fluids generally show higher D values for Zr and Hf compared to silicic melts[80], which may explain the minor mismatch in our numerical models involving silicic HDFs on Fig. 9c. Unfortunately, no incompatible trace element volatile data are currently available for eclogite-derived supercritical fluids and modelling their interaction with depleted cratonic mantle and the $H_2O$ and F content of the orthopyroxene-rich garnet harzburgites is not possible. More robust modelling also requires experimentally determined D values for silicic fluids in equilibrium with garnet (including volatiles) at relevant pressure and temperature conditions.

## Carbonation, hydration, fluorination and chlorination of cratonic mantle

Our research on orthopyroxene-rich garnet harzburgites expands on previous studies by demonstrating that the formation of excess high Mg# orthopyroxene was accompanied by an increase in the carbonation, hydration, fluorination and chlorination of the cratonic mantle. Our preferred model for the formation of excess orthopyroxene—along with garnet, phlogopite and occasionally diamond—involves interactions between extensively melted peridotite (produced during adiabatic decompression at shallow depths) and supercritical fluids derived from down-going slabs of oceanic lithosphere (Fig. 10). The behaviour of fluorine in supercritical fluids is less well understood than H and C, but it is known that serpentinites contain high concentrations of F (up to 400 μg/g) and release it, along with $H_2O$, sulfur and chlorine, during prograde metamorphism of subducting oceanic slabs[81]. We suggest that serpentinite-derived fluids will have variable $H_2O$/F ratios but low concentrations of REEs, low Ce/Pb and high MgO[82]. They may infiltrate overlying lithologies, which, if carbonated, will potentially release supercritical C-O-H-F-Cl fluids as the slab descends. These supercritical fluids will be highly mobile and reactive, and may be released at depths up to 300 km[83].

A major obstacle for the involvement of slab-derived silicic melts and fluids in the formation of orthopyroxene-rich harzburgites has been the similarity of their oxygen isotope ratios ($\delta^{18}O$ = + 5.74 ± 0.27 ‰) and other cratonic mantle peridotites to mid-ocean ridge basalts (MORB)[18,24]. This has been interpreted by some as evidence that the protolith of the silicic agents did not experience seafloor serpentinization. However, serpentinization at higher temperatures and deeper in the oceanic crust may generate $\delta^{18}O$ of +4 to +6 ‰[84]. Moreover, preservation of mantle-like $\delta^{18}O$ values may result from low fluid-rock ratios and buffering of the infiltrating fluid composition to peridotite values. This may explain why mantle peridotites from the Colorado Plateau (SW USA), which have trace element signatures characteristic of subduction-related metasomatism, generally show $\delta^{18}O$ values that are similar to those of orthopyroxene-rich harzburgites and other cratonic mantle peridotites and eclogites[85]. As a result, $\delta^{18}O$ values may not be a fully reliable indicator of slab-derived fluids from subducted oceanic lithosphere, and more internally consistent datasets are needed to evaluate this hypothesis.

Our work shows that modification of Archaean melt residues occurred in the continental mantle by the reactive infiltration of a silicic and Mg-bearing supercritical fluid with high C, Pb, low Sr and $H_2O$/F. The reaction of these supercritical fluids with cratonic mantle olivine would cause the formation of excess high-Mg# orthopyroxene and account for the observed elevated Fo contents, high bulk $H_2O$, low Mg/Si ratios, depleted Ti levels and positive HREE slopes that are characteristic of orthopyroxene-rich garnet harzburgites found in xenolith suites from several global cratons. At high pressures (>~5 GPa) in reduced cratonic mantle, the reactive infiltration and cooling of fluids with high C-O-H contents may lead to diamond formation[43]. Additionally, these fluids are rich in potassium and may react with orthopyroxene in the harzburgite to form phlogopite[26]. The

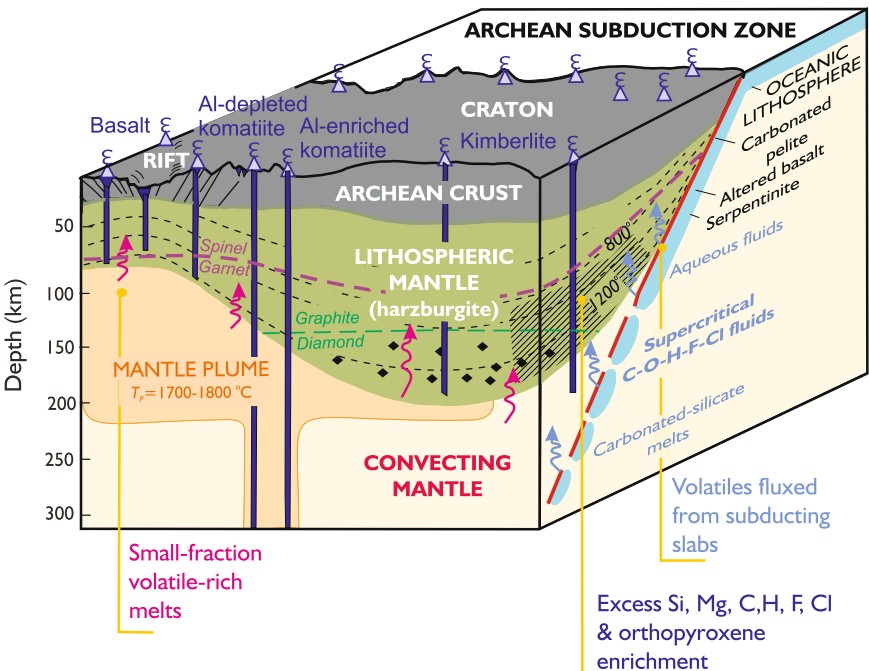

**Fig. 10 | Schematic illustration of the formation of excess orthopyroxene in Archaean cratons.** Hashed region illustrates where orthopyroxene-rich garnet harzburgites are most likely to form in relation to subduction zones. Devolatilisation of carbonated pelite, altered oceanic crust or serpentinite at subduction zones −via aqueous fluids, supercritical fluids or carbonated-silicate melts−provides the primary flux of C, H and halogens (F, Cl) into the convecting mantle[73,105–108]. Fluxing of the carbonated pelites by fluids from underlying basaltic crust (eclogites) and serpentinites may lead to the mobilisation, ascent and reactive infiltration of Si, Mg, C, F, $H_2O$ and Cl in the overlying lithospheric mantle to form excess orthopyroxene and volatile enrichment. Since the Archaean, some of this signature may have been overprinted by infiltrating small-fraction volatile-rich melts from the convecting mantle, especially at the base of the lithosphere. The diamond-graphite boundary is from ref. 109.

experimental study of Gao et al.[25] involving Si-rich melt with orthopyroxene-poor harzburgite suggests that excess orthopyroxene and phlogopite may form at temperatures between 1100 and 1200 °C, and we anticipate lower temperatures for silicic fluids. The slightly lower Ba levels in the observed data compared to our models may reflect incomplete phengite breakdown in the down-going slab at pressures and temperatures below the stability limit.

**Orthopyroxene-rich garnet harzburgites, C-O-H-F-Cl fluids and diamonds**

C-O-H fluids have been implicated in the formation of diamond-hosted pyrope garnets with sinusoidal chondrite-normalised REE patterns that peak at Sm and Nd[45] and are similar to those in the orthopyroxene-rich harzburgites (Supplementary Fig. 7)[5,15,19,86]. Moreover, olivine inclusions found in diamond with a harzburgitic paragenesis (based on the composition of co-existing garnet) from Bultfontein, De Beers, Dutoitspan and Wesselton on the Kaapvaal craton also have very high Mg# (94.5 to 97.5)[86], with a range similar to the most magnesian olivines in the orthopyroxene-rich garnet harzburgites (Supplementary Fig. 4).

Although chlorine is a major component in brines trapped within diamonds, it is highly incompatible in olivine and pyroxenes[87] so that during reactive infiltration of the lithospheric mantle, chlorine will become concentrated in metasomatic fluids until it reacts to form accessory minerals such as phlogopite, amphibole, and apatite. While phlogopite is rarely reported from diamond inclusions, we note that those found associated with a harzburgitic paragenesis may also contain high Cl and $Cr_2O_3$ but low $TiO_2$ contents[88] and similar to those in the orthopyroxene-rich garnet harzburgites (Supplementary Data 4). The presence of phlogopite with relatively high Na contents in the most orthopyroxene-rich garnet harzburgites is evidence of the alkaline nature of the percolating silicic fluids.

The presence of diamond in a Kaapvaal orthopyroxene-rich garnet harzburgite suggests that silica enrichment occurred after cratonisation. We conclude that the most likely mechanism for explaining the volatile content of cratonic mantle with excess orthopyroxene and silica is through C-O-H-F-Cl fluids originating from subducting slabs (Fig. 10). Notably, in the Kaapvaal mantle, the F content in both olivine and pyroxenes in orthopyroxene-rich garnet harzburgites seems to increase with the amount of excess orthopyroxene (as shown in Fig. 7) and the occurrence of diamond. We infer from this that the increased volume of C-O-H-F fluid promotes a greater interconnected network along grain boundaries, which allows more extensive reactive infiltration and diffusive exchange of Mg, Si, K, C, $H_2O$ and F with both pre-existing and new mineral phases. The presence of diamond in orthopyroxene-rich garnet harzburgite sample BD2125, which last equilibrated at 1055 °C and 4.4 GPa[10], suggests it came from a region more reduced than the enstatite-magnesite-olivine-diamond (EMOD) buffer. We propose that under these reducing conditions, fluorine is incorporated into olivine (instead of forming OH groups) and gives rise to the F-rich bulk composition of BD2125. Additionally, our work suggests that Si-rich C-O-H-F-Cl supercritical fluids may have a higher $H_2O$/F ratio than those that are less silicic but more Mg-rich and form dunites (e.g., BD1153), but more work is required to confirm this.

Our findings align with previous studies that highlighted the common association between diamond and harzburgitic garnet inclusions[43]. We also note that olivine inclusions within diamonds sometimes have high F concentrations and low $H_2O$ levels[61,89], though it is unclear whether these inclusions formed alongside or before the host diamond[90]. Additionally, our interpretations are consistent with those that have shown the formation of diamonds in harzburgites primarily occurred in the PalaeoArchean, driven by subduction-related fluids[91]. Further support for the idea of recycling of subducted material

in the Archaean beneath the Kaapvaal craton is provided by sulfur isotope ratios of sulfide inclusions in diamond[92] together with nitrogen and carbon isotope data, which indicate that sedimentary carbon was being recycled >3 Ga[93] and plate tectonic processes similar to those at the present day were operational.

## Broader implications

While our study has primarily focused on the association of volatiles and orthopyroxene-enrichment in the Kaapvaal craton mantle, where some of the greatest excess orthopyroxene and silica have been observed, similar mantle peridotites have been widely reported from the Siberian, Slave, Rae and Tanzanian cratons[5,9–17]. There are, however, no F or Cl data for olivine and pyroxenes in well-characterised peridotites from the Siberian, Slave and Rae cratons and systematic analytical work is now required to firmly establish if the enrichment in multiple volatiles and orthopyroxene is a global phenomenon. This has broader implications for Archaean tectonics and also the long-term stability of continental cores because widespread inter-actions between subduction-related supercritical fluids (derived from serpentinite, carbonated eclogite and sediments) and existing minerals may modify the rheology of cratonic mantle and decrease the viscosity contrast with the underlying convecting mantle[10]. A striking example of this is the North China craton, which experienced major thinning in its eastern part due to the Mesozoic influx of subduction-related fluids[94]. Nevertheless, the extent to which slab-derived, supercritical C-O-H-F-Cl fluids may have impacted the strength of other continental roots, or why the North China craton has been so significantly modified, remains unclear. Our research also underscores the critical need for further high-temperature and high-pressure experiments involving fluorine- and C-O-H fluids. These experiments are vital to unravel the partitioning behaviour of multiple volatiles, shedding new light on the formation, evolution and stability of Earth's continental cores.

## Methods

### Samples

Our micro-analytical study has focused on olivine and pyroxenes, which are known to contain variable but potentially important con-centrations (up to 100's of μg/g) of H[49,95,96] and F, and low amounts of Cl[1,10,50,52,53]. These are from well-characterised peridotite xenoliths from the Archaean Kaapvaal (Mothae and Bultfontein; Fig. S1) and Tanzania cratons (Lashaine) and, for comparison, from post-Archaean con-tinental off-craton locations (SW USA, Mongolia, Antarctic Peninsula and the Eifel, W. Germany). The thin sections were digitally scanned, and modal abundances of mineral phases were determined using the JMicroVision image analysis toolbox (www.jmicrovision.com/index.htm, version 1.2.7). A minimum of 500 points was undertaken for each scan. The harzburgites, lherzolites and dunite analysed in our study represent the full spectrum of peridotite lithologies found in global on-and off-craton settings, cover a large pressure (2 to 6 GPa) and tem-perature range (800–1270 °C) and include a rare diamondiferous harzburgite (BD2125, Supplementary Data 1). This coarse-grained harzburgite was the first mantle peridotite in which in situ diamonds were reported in the published literature[39] and exhibits the most extreme enrichment in orthopyroxene (39%) and phlogopite (1.2 modal%). The compositions of minerals in the orthopyroxene-rich garnet harzburgites analysed in our study span the full range observed in this lithology in global cratons, with olivines varying from $Fo_{91.8-93}$ (Fig. S2) and orthopyroxenes with Mg# between 92.9 and 93.9[9,18].

### Analytical methods

Prior to analysis, all of the mantle xenoliths described in this study were first examined in thin section to ensure that they were petro-graphically fresh and free of serpentinisation. Individual mineral grains were carefully hand-picked from the crushate. Only grains free from melt and mineral inclusions were selected for cleaning, mounting with crystal bond, and then polished to a 1 μm finish in the clean laboratory at the Department of Earth Sciences, University of Cambridge. Both samples and polishing pads were regularly cleaned in an ultrasonic bath to minimise contamination. The polished mounts were examined using an SEM under low vacuum conditions to verify the selection of the correct mineral phases. Once verified, the polished crystals were removed from the temporary crystal bond, cleaned with acetone and mounted in indium within aluminium disks.

### Mineral major and minor element chemistry

In situ analysis of major and minor elements in olivine, clinopyroxene, orthopyroxene, garnet and phlogopite was conducted using the Cameca SX-100 electron microprobe analyser (EMPA) at the Depart-ment of Earth Sciences, University of Cambridge. The instrument is equipped with five WDS detectors and one EDS detector. The analysis was performed using an accelerating voltage of 15 keV, a beam current of 20 nA, and a beam size of 1 μm. The counting times varied, with 10 s for Na and K (alkali elements), 20–30 s for major elements, and up to 140 s for minor elements to minimise error. A higher current (20 nA) was employed to enhance the number of X-rays detected and lower the detection limits. Calibration standards included Na on jadeite, K on K-feldspar, Fe on fayalite, Mn, Ni, and Cr on pure metals, Si and Ca on diopside, Al on corundum, Mg on St Johns olivine, and Ti on rutile. Secondary standards, including San Carlos olivine, augite, diopside, and basaltic glass, were analysed before, during, and after each ana-lytical session. The ZAF correction method was used to adjust for matrix effects.

### Mineral trace element chemistry

In situ trace element analysis was undertaken using an ESI UP193NC laser ablation system attached to a Nexion 350D ICP-MS system (LA-ICP-MS) in the Department of Earth Sciences at the University of Cambridge. The NWR193 is equipped with a TwoVol2 sample chamber and small volume sample cup to quickly and efficiently transfer sam-ples and washout performance. The analysis set up involved a 100 μm diameter laser beam, laser repetition rate of 10 Hz for clinopyroxene and garnet and 20 Hz for olivine and orthopyroxene, and laser fluence of 8 J cm$^{-2}$ set at 50% power in order to ensure optimum signal intensity while minimising down hole fractionation. The LA-ICP-MS data acqui-sition settings were 1 sweep per reading, 80 readings, 1 replicate, and the total data acquisition lasted 60 s (approximately 1 data point for each element per second). The laser was programmed to run for 40 s on clinopyroxene and garnet, and 35 s on olivine, with a laser warm-up time of 20 s for each spot analysis in both instances.

During LA-ICP-MS analysis, the spots were carefully chosen to avoid contamination from nearby cracks, inclusions, or any over-lapping neighbouring phases. Ablated sample material was carried in pure helium gas at 700 mL min$^{-1}$, which was joined with the argon nebulised gas line from the ICP-MS using a signal homogenised device at 0.8 L min$^{-1}$ before the ICP-MS torch. The ICP-MS dwell time for each mass was dependent on the isotope and concentration of the element in the samples, but was typically 20 milliseconds (ms) for most trace elements in clinopyroxene and garnet, and this was increased up to 100 ms for some low concentration elements in olivine. Clinopyroxene and garnets were analysed using CaO and $SiO_2$ from EMPA analysis as internal standard normalisation. NIST612 was used to calibrate ele-ment concentrations. Accuracy and drift were assessed using NIST 614, while BCR-2G, BIR-1, BHVO-2, GOR132-G and in-house clinopyroxene and garnet standards were also analysed. The results for these stan-dards are reported in Supplementary Data 2 and Jackson and Gibson[10]. Clinopyroxenes were re-analysed in October 2023 so that the dataset included Ba, Rb, Th and U−these are absent in the dataset of ref. 10. Data was processed using Glitter software (GEMOC, Australia). The data were filtered for background and melt/mineral inclusions after

each analysis to optimise the data. Replicate spots were then averaged. Accuracy is good for data, with results typically between 80 and 110%.

### Analysis of H, F and Cl by SIMS

Concentrations of H, F and Cl in olivine and pyroxenes were determined by Secondary-Ion Mass Spectrometry (SIMS) at the NERC Ion Microprobe Facility at the University of Edinburgh. SIMS analysis was undertaken before EMPA and LA-ICP-MS analysis to avoid contamination and elemental migration under the electron beam. A Cameca IMS 4f was used for Kaapvaal samples analysed in September 2018 and May 2019, and a Cameca IMS-7f GEO for samples from Tanzania and off-craton locations analysed in November 2019 and May/June 2021.

Crystal mounts were gold-coated before being loaded into the vacuum chamber of the SIMS instrument for over 48 h (often in excess of 60 h) before the first analysis. An $O^-$ primary beam was used with a 10 kV primary accelerating voltage and 5 nA current. Prior to each analysis, an area of 50 μm (Cameca IMS 4 f) or 15 μm (Cameca IMS-7f GEO) was chosen to raster for 4 min. The acquisition went through 10 (Cameca IMS 4 f) or 20 cycles (Cameca IMS-7f GEO), but only the last 50% were used. An energy offset of 75 was chosen to reduce the interference without losing signal. Prior to each analysis, the peaks for Si, Mg, Ca, H, F and Cl were aligned. Si was used as the reference and in ratios to calculate other concentrations. Mg and Ca were analysed for cross-referencing.

At the start of each day, background values were measured by analysing the anhydrous olivine reference standards OL DC212, OL116610-5, Ol-Koh and Ol-Fo. The concentrations of $H_2O$ and F in the mineral separates were calculated using a calibration line based on the ratio of H/Si counts for standards with known $H_2O$ and F concentrations. Clinopyroxene and orthopyroxene reference standards with known $H_2O$ and F concentrations (CPX-SC-J1, CPX-KH03-4, CPX-SMC31139-1, CPX SMC31011, OPX KH03-4, OPX 116610-3; Ref. 97) were used for the calibration. Concentrations of H and F in olivine were calculated using the basaltic glass standard ALV-519-4-1 (Supplementary Data 3) and then pyroxene standards because the matrix effect is established to be minimal (C.J. De Hoog, pers. comm). San Carlos olivine was repeatedly analysed throughout each run, as it has low concentrations of $H_2O$ (e.g., Aubaud et al.[98]) and serves as an internal reference to track any analytical drift (Supplementary Data 3).

Chlorine concentrations were not available for any pyroxene or olivine reference standards during the analysis of mineral grains from the Kaapvaal craton (Mothae and Bultfontein) in September 2018. The raw data were reprocessed through JCION5 using a basaltic glass, ST8-1A9, with known Cl concentration (2025 ppm; Supplementary Data 3). The counts were then normalised to the Si concentrations from prior EPMA and Cl concentrations calculated using the ST8-1A9 composition and the conversion factor established using Si counts. For mineral grains from off-craton peridotites and also for Lashaine in Tanzania concentrations of Cl were calculated using ST8.1.A92-2. The Cl acquisition technique was later refined to include optical glass herasil, containing 0.4 μg/g Cl, for background monitoring, which was generously provided by B. Urann. This consistently showed low Cl backgrounds, ranging from 0.3 to 0.6 μg/g.

The raw data were initially processed by C Talavera Rodriguez using the JCION5 software, followed by manual background correction on Microsoft Excel. $H_2O$ and F concentrations of the mineral separates were calculated from a calibration line of the ratio of H/Si counts for standards with known $H_2O$ and F concentration (Supplementary Data 1). The error for all $H_2O$, F and Cl concentrations is calculated by summing the instrument error and the standard deviation in point analyses averaged for each crystal (Supplementary Data 3). The instrument error is calculated by taking the square root of the sum of

the square of the % count error in H and Si, F and Si or Cl and Si.

$$\% \text{ instrument error} = \sqrt{a(\% \text{ error}_{Si})^2 + (\% \text{ error}_H)^2}$$

Mineral grains in two samples of Kaapvaal peridotites were subsequently analysed for $H_2O$ by FTIR[99] and revealed comparable results (Supplementary Data 3).

## Data availability

The authors declare that the data supporting the findings of this study are available within the paper and its supplementary information files.

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

## Acknowledgements

Samples of mantle peridotites analysed in this study are from the JB Dawson and D McKenzie collections curated by S.A.G. in the Department of Earth Sciences at the University of Cambridge. We are grateful to Iris Buisman and Jason Day at the University of Cambridge and Christina Talavera at the University of Edinburgh for their assistance with EPMA, LA-ICP-MS and ion probe analyses, respectively. C.G.J. and J.C. were funded by Natural Environment Research Council studentship grant NE/L002507/1, and S.A.G.'s research on volatiles by NE/Y000218/1.

## Author contributions

S.A.G.: Conception of the work, writing, drafting of figures, numerical modelling and editing. C.G.J.: Characterisation and geochemical analysis of xenoliths from the Kaapvaal craton. J.C.C.: Characterisation and geochemical analysis of xenoliths from Tanzania craton and off-craton locations. J.A.F.D.: Geochemical analysis of xenoliths from the Kaapvaal craton.

## Competing interests

The authors declare no competing interests.
