## [Transparent Peer Review file · Nature Communications]

The role of C-O-H-F-Cl fluids in the making of Earth's continental roots

Corresponding Author: Professor Sally Gibson

A version of this paper was originally rejected for publication by Nature Communications, however that decision was reconsidered after appeal by the authors.

Version 0:

Reviewer comments:

Reviewer #1

(Remarks to the Author)

Gibson et al. presented the volatile (H₂O, F, Cl) contents in Kaapvaal Craton peridotites to decode the origin of orthopyroxene (Opx) enrichment in the cratonic lithospheric mantle. Their results indicate that the mean bulk H₂O content of the orthopyroxene-rich harzburgites is similar to that of garnet harzburgites and spinel harzburgites, but higher than that of garnet lherzolites. In contrast, the mean bulk F contents are lower in the orthopyroxene-rich garnet harzburgites than in both the garnet harzburgites and spinel harzburgites, but higher than in the garnet lherzolites. These variations are interpreted to result from reactions with supercritical C-O-H fluids, rich in silica, fluorine, and chlorine, that infiltrated from subducted oceanic lithosphere during the Archean. Given the new and important volatile data presented for these peridotites, I recommend publication after revisions, considering the following suggestions and comments:

The supercritical C-O-H fluids from serpentinites, which infiltrated through carbonated pelites, should also be enriched in Na, K, and P, not just Si. Did the authors examine major element variations in relation to the Opx enrichments? Moreover, significant Opx enrichment would require modal metasomatism, which implies that whole rock trace elements might have been reset by supercritical fluids, especially given the refractory nature of harzburgites with low trace element contents. These Opx-rich harzburgites do not show LREE enrichments, which would be expected from the supercritical fluids infiltrating sediments (see D from Kessel et al. 2005, Nature). A mixing calculation between refractory harzburgites and supercritical C-O-H fluids equilibrated with sediments should be performed.

Generally, F compatibility in mantle silicate minerals follows the order: garnet > pyroxene > olivine. Interestingly, the Kaapvaal Craton Opx-rich harzburgites show the opposite trend, with olivine exhibiting much higher F contents than pyroxenes. The authors need offer an explanation for this observation.

Fluids are generally enriched in Cl relative to F, as also suggested by Fig. S7 in the manuscript. Is it possible to reconstruct the bulk Cl contents to further investigate the role of supercritical C-O-H fluids? I also noticed some discussion about alkalinity addition through diamondiferous orthopyroxene-rich garnet harzburgites, but more details are more helpful.

The methods section provides detailed information on volatile measurements using SIMS, but it does not address the reliability of the results. Some standard references as blind samples should be used to estimate the reliability of their results.

I hope these suggestions help improve the manuscript.

Minor comments

Line 823. elements to element

Line 853. % to wt.%

Line 872. What are the residues predicts? Melting residues?

Line 888. MORB source normalized?

Line 913. is from Ref. 100

(Remarks to the Author)

The role of C-O-H-F-Cl fluids in the making of Earth's continental roots

This manuscript by Gibson et al. addresses a longstanding and significant open question in mantle geochemistry regarding the origin of Si enrichment observed in some of Earth's cratonic 'roots.' Since Boyd's initial observations in 1989, this phenomenon has been the focus of numerous studies, leading to two primary schools of thought: 1. orthopyroxene enrichment is attributed to be a key characteristic inherited from a nonpyrolytic mantle, or 2. Secondly, silicon enrichment in the form of excess orthopyroxene (opx) occurs in preexisting olivine-rich melting residues, such as metasomatism in subduction zones or plume settings. Overall, the entire manuscript is a pleasure to read and very clear.

The authors tackle this question by analyzing a suite of mantle xenoliths and their minerals for volatile contents, including H₂O, Cl, and F, with a particular focus on the opx-rich xenoliths from the Kaapvaal Craton, including one diamond-bearing opx-rich harzburgite. The authors argue that based on the observed volatile enrichment of olivines within the opx-rich xenoliths, the most likely cause of the opx enrichment was the infiltration of a small volume of supercritical C-O-H-F-Cl-bearing fluids sourced from subducted oceanic lithosphere ± sediment and its reaction with a previously depleted mantle to form the observed Si enrichment. The above is in contrast to recent thermodynamic studies, which propose a reaction between "dry" komatiitic melts and a melt-depleted lithospheric mantle at high pressure, forming opx in a peritectic reaction (e.g., Tomlinson and Kamber 2021).

The Cambridge group is one of the pioneering teams in the field and is an expert in the analysis of volatiles in mantle minerals and their storage at lithospheric conditions. This large body of work on the volatile concentrations (F, H₂O, and Cl) on 22 xenoliths and 65 mineral separates coupled with petrographic major and trace element analysis represents a very impressive analytical campaign and achievement. In my opinion, this contribution is vital to the community and keeps the long-lasting debate alive.

After some moderate to significant revisions, this study should be published in Nature Communications.

Major remark

It appears that the entire model is heavily reliant on a single diamond-bearing, opx-rich harzburgite with approximately 120 ppm F, along with two additional xenoliths, all from the Bultfontein locality. While the scarcity of suitable samples is understandable, drawing robust correlations from only three data points is challenging. Additionally, making a global assessment based on such limited data, which primarily pertains to the Kaapvaal Craton, seems to be overreaching. Given the long residence time of these xenoliths in the mantle, a more detailed explanation is needed to clarify how the samples retained such high F contents over geological timescales.

Specific remarks in the text

Abstract

The abstract is well written very lays out the question very elegantly, and the novelty of the study.

Line 13: Why mention Siberia and Rae and Tasmania cratons here? It is a bit misleading in my mind.

Line 14: What is the significance of the Ti depletion? Maybe highlight this observation a bit more here?

Line 20: A bit vague, in my opinion.

Introduction

Line 48: Mg# notation is explained in line 50. I would move it here.

Results

Major and trace element chemistry

Line 110: What is the significance of phlogiopite? Is this phase related to kimberlite metasomatism, or is it more ancient?

Volatile elements

It would be helpful to visualize the different populations and how they relate to one another, perhaps using a box-and-whisker diagram similar to the one in the supplemental file. Additionally, many of the measurement error bars appear to overlap. If this is typical in the field of volatiles in the mantle, an explanation of why the results remain robust would be beneficial. For example, in lines 144–146, there is a clear population difference, and this should be represented in a figure to strengthen the argument.

Bulk volatile content

Line 185: How robust are the correlations that are not shown? This seems like a critical point to address. Additionally, are there correlations with other minor or trace elements in olivine and orthopyroxene? It might be interesting to examine first-row transition elements such as Zn, Sc, V, and Mn.

The current projection in Figure 5, illustrating the relationship between bulk LREE versus F and the Sm/Nd peak position, could be improved, as it appears somewhat misleading. Alternative projections, such as plotting \sum LREE (ppm) versus F (ppm), Si, Mg, or Nd/La versus F, Si, Mg, might provide clearer insights. It might also be worth considering a more integrated presentation of panels A and B to better highlight the differences between the opx-rich and opx-poor samples.

Mantle Melting Models

I think this and subsequent sections are part of the discussion, but it is not noted in the text.

Line 193: I believe you meant 2b and not 2a? Additionally, I found the use of the color gradient a bit excessive. If there is a strong correlation, it might be worth dedicating its own panel—just a suggestion. Line 192-193: Fo. Molar proportions, yes?

Reactive infiltration of Al-enriched komatiites

My main comment here is that the proposed fluid-associated opx addition is comparable to melt addition, except for one sample that is a bit strange and needs to be addressed. The Ti and HREE mismatch is very interesting.

Infiltration of subduction-related melts and fluids

Line 249-252 I think a citation is needed for this statement.

Why did you choose the fluid composition of Sokol et al.? ~1100°C at ~8 GPa seems a bit cold for Archean tectonics. What geotherm does this plot on? It might be worth looking at the Timmerman et al. (2022) EPSL paper on Mesoarchean diamonds to see if their thermal modeling could explain the cold and transient conditions relevant to your proposed fluids.

Is there additional evidence supporting monazite dissolution? Including references to radiogenic isotope studies could strengthen this argument. The explanation for subaerial weathering seems a bit too convenient.

It might also be valuable to explore natural examples of silicic fluids trapped in diamonds as potential candidates for the required supercritical fluid. The recent compilation by the HUI group on fluid compositions in diamonds could be relevant. While there is no F content data for these inclusions due to hydrofluoric acid use during sample prep, H₂O, Cl, Si, Al, and other trace elements are well-characterized and are known metasomatic agents.

Regarding diamond-forming fluids, to my knowledge, there are no fibrous diamonds from the Kaapvaal of this age that are SiO₂-rich. However, their absence in the diamond record does not necessarily exclude their involvement. The oldest observed fluids in the Kaapvaal are carbonatitic, which cannot account for Si enrichment. You might consider looking at Weiss et al. (2021, Nature Communications) on how fibrous diamonds track the metasomatic history of the Kaapvaal using helium isotope systematics.

The "smoking gun," in my view, would be high-Mg olivine in diamonds with high F, Cl, and H₂O. While this might be beyond the scope of this study, it is worth considering. On that note, olivine inclusions from the De-Beers pool kimberlites have a distinct high-Mg number and could provide valuable insights (see Stachel et al., 2022, RIMG review, Figure 6).

I'm not sure that the segway to the next section is warranted. Maybe lines 279-300 should be incorporated here, and sections Infiltration of subduction-related melts and fluids & Carbonation, hydration, fluorination, and chlorination of cratonic mantle should be renamed and restructured.

Line 289: The major obstacle is not just oxygen isotopes but also mass balance considerations and the cold geotherm that you just noted above.

Line 329: The Slave and Kaapvaal cratons have very different geological histories. Before reaching this conclusion, Slave xenoliths should be added.

Line 344: Why is this seen in Fig S3? Maybe you mean S4? If so, I think this (or a version of it) should be moved to the main text.

Line 344-347: This claim warrants a figure in my mind; also, it is worth noting that the K and (F) in K-rich phases are likely lost during subsequent melting.

Lines 364-376: The link to wider implications is very interesting but a bit of an overpromise and underdeliver, especially for the North China Craton. Is there any insight as to why the opx enrichment is restricted to the Kaapvaal Slave and Siberian Cratons?

Some other remarks

The experimental results of Gao et al. (2021) might be worth considering aside from the introduction as they are similar to what the authors propose.

What about using the $\Delta\text{Mg}/\text{Si}$ notation of Canil and Lee 2009? You did something similar in supp Fig. S4 but did not add volatiles to the color bar. It seems from Fig. 2 that there are no robust correlations. Don't you expect that the more the xenolith departs from the Boyd trend, the more fluid reacts, and thus, the more F in the olivine? Here, some diffusion modeling might come in handy.

Why does BD1153 (dunite) have so much F in it?

Figures and supplementary materials remarks

The Figures overall are very engaging. I have a couple of remarks and possible suggestions.

Fig. 1: Since the focus is primarily on the Kaapvaal Craton, could you include a panel showing the location of the retrieved xenoliths? Or provide it in the supplement? Is there any observed spatial variation?

Fig. 2 Panel A might be redundant; I think that Panel B is the star of this figure.

Fig. 3 See comment above. Plenty of projections might be better utilized to illustrate the differences you are discussing.

Why normalize to MORB source and not Primitive mantle? From the way I read the manuscript, it is not clear why you chose to show this normalization.

Fig. 5. Are there any inclusions in the diamond to plot on this diagram? Why don't you show the olivine from the opx-rich sample in panel a? It is a bit misleading. In panel b, the diamond-bearing opx-rich sample BD2125 has 120 ppm F in the supplementary table and is plotted as ~138 ppm in the figure. Am I missing something? Is there any meaning to the correlations?

Figure 8 is great, but it has been recycled from other publications. If you want to show a conceptual model, I believe it merits a new figure.

The supplemental tables need a bit more work to make them clear to the reader. missing descriptions

In the supplementary table of the bulk trace element compositions, there is no table description indicating which samples are from the Kaapvaal, Tanzania, or off-craton. For those who are not familiar with the local names, it is hard to follow. Also, can you group them in a systematic order?

Supp table Bulk trace element compositions There is data in columns O and P that should be deleted.

The clinopyroxene table needs a description.

I have no remarks on the Methods section

Best,

Oded Elazar

Reviewer #3

(Remarks to the Author)

Review of the manuscript "The role of C-O-H-F-Cl fluids in the making of Earth's continental roots" by Gibson, Jackson, Crosby, and Day, submitted to Nature Communications

The focus of this manuscript is the existence of opx-rich garnet harzburgites in the SCML.

The authors examine 3 scenarios for the formation of excess opx in these xenoliths: high-pressure melting residues,

komatiite interactions, or subduction-related silicic melts and fluids. The authors analyze olivine, pyroxene and garnet in different xenoliths for H₂O, F and Cl content and conclude that the best model is the interaction of the harzburgitic mantle with C-H₂O-F-Cl fluids derived from dehydration of the subducting slab.

The formation of opx-rich harzburgites is a long-standing question. The suggestion that these xenoliths were formed due to the interaction of depleted harzburgite with slab-derived silica-rich fluids or melts is not new. However, this manuscript attempts to provide additional support for this model via the analyses of volatiles in the peridotite minerals themselves, and provides unique information for the type of fluid/melt interacting with the harzburgite (volatile-rich). This manuscript is important and very interesting.

The manuscript is well written and easy to follow.

Major comments and thought

1. Opx-rich harzburgites are mainly found in Slave, Rae, Kaapvaal and Siberia cratons. Does that mean that fluids interacted with the SCML only in these regions and no other SCML?
2. There is no discussion on carbon in these potential fluids.
3. Saline HDF (found in fluid inclusions in diamonds) were identified in many cratons around the world (see review by Weiss et al. 2022). Not just were there is excess opx. Also, Weiss and others suggested that silicic HDFs originate from the dehydration of the subducting slab. It will be interesting to use the trace element compositions of either the saline HDF or silicic HDF instead of experimental supercritical fluid in Figure 7.
4. Elazar and Kessel (2024) interacted harzburgite with eclogite in a hydrous environment (CHO fluid) at the conditions above the eclogitic wet solidus, but below the wet peridotite solidus. At the experimental conditions, eclogitic supercritical fluid was interacting with the harzburgite. During this interaction, the amount of ol decreases while the amount of opx increases. The olivine and opx Mg# decrease due to such an interaction. Their work may apply to the suggested model.

Minor comments

5. Line 90 - The authors use Kessel et al. (2005) (ref. 35) for trace elements profiles during enrichments. Kessel et al. (2005) focuses on trace elements in eclogite+H₂O system. It is better to refer to Kessel et al. (2015): The behaviour of incompatible elements during hydrous melting of metasomatized peridotite at 4–6 GPa and 1000 °C–1200 °C.
6. Line 116 – REE profiles are in Figure S5 (not S4).
7. Lines 131 - Please check the calculated averages for the H₂O and F in the various minerals and in the bulk peridotites. Following the data presented in Table S1, I calculate different values. For example, The H₂O content in olivine in opx-rich garnet harzburgite in Table S1 are 37, 69 and 69 ppm. The average is 58±18 ppm (and not 49.5±68 as states in the text). Same for opx etc.
8. Line 191-193 – Figure 2a does not show bulk H₂O+F content. Figures 2b does. The authors declare that the Phanerozoic oceanic melt residues show a decrease in bulk H₂O+F with increasing Fo content and model abundance of ol.
9. Line 236-237 – Ti seems to fit well with the infiltration of Al-enriched komatiites model. Ba and Rb do not fit.
10. Figure 7 – In the legend, the reference is Sokol et al (2024). In the figure caption, it is Sokol et al. (2023).
11. The fluid taken from this reference is a supercritical fluid in equilibrium with a pelite. Please explain this choice. Why not choose a supercritical fluid in equilibrium with eclogite
12. Check the symbol of the supercritical fluid from Sokol in Fig. 7b. (Triangle in the legend, circles in the figure).

Version 1:

Reviewer comments:

Reviewer #1

(Remarks to the Author)

Thank you to the authors for addressing my concerns. I am happy to recommend this version for publication, subject to the following minor suggestions:

I suggest changing "continental roots" in the title to "cratonic roots" for greater precision.

Line 23: Please specify the type of volatiles being referred to.

Line 135: Please note the lowest Ce/Pb ratios observed.

Reviewer #2

(Remarks to the Author)

The authors have addressed all of my previous remarks, and the revised manuscript looks excellent!

Just one thought that came to mind during my second read (and apologies if I missed this)

Do you have any nitrogen aggregation data on the diamonds hosted in the diamondiferous orthopyroxene-rich garnet harzburgite (BD2125)?

Although previous work has clearly shown that these diamonds are old, it would be a nice addition if you have nitrogen aggregation systematics—especially if the diamonds have high nitrogen contents and show significant aggregation. This could further strengthen the interpretation of an ancient subduction-related origin.

I also took a moment to look for any remaining typos, and everything looks good to me. I trust the editorial team will give it one final check before publication.

Other than that, the paper looks great!

All the best,
Oded Elazar

Reviewer #3

(Remarks to the Author)
see attached file

RESPONSE TO REVIEWERS' COMMENTS

Reviewer #1 (Remarks to the Author):-

Gibson et al. presented the volatile (H₂O, F, Cl) contents in Kaapvaal Craton peridotites to decode the origin of orthopyroxene (Opx) enrichment in the cratonic lithospheric mantle. Their results indicate that the mean bulk H₂O content of the orthopyroxene-rich harzburgites is similar to that of garnet harzburgites and spinel harzburgites, but higher than that of garnet lherzolites. In contrast, the mean bulk F contents are lower in the orthopyroxene-rich garnet harzburgites than in both the garnet harzburgites and spinel harzburgites, but higher than in the garnet lherzolites. These variations are interpreted to result from reactions with supercritical C-O-H fluids, rich in silica, fluorine, and chlorine, that infiltrated from subducted oceanic lithosphere during the Archean. Given the new and important volatile data presented for these peridotites, I recommend publication after revisions, considering the following suggestions and comments:

- 1) The supercritical C-O-H fluids from serpentinites, which infiltrated through carbonated pelites, should also be enriched in Na, K, and P, not just Si. Did the authors examine major element variations in relation to the Opx enrichments? *We observed an enrichment in K (but not Na) and do not have data for P. As mentioned in the text, the K enrichment is linked to phlogopite.*
- 2) Moreover, significant Opx enrichment would require modal metasomatism, which implies that whole rock trace elements might have been reset by supercritical fluids, especially given the refractory nature of harzburgites with low trace element contents. *Yes, we fully agree with this statement in our manuscript.*
- 3) These Opx-rich harzburgites do not show LREE enrichments, which would be expected from the supercritical fluids infiltrating sediments (see D from Kessel et al. 2005, Nature). A mixing calculation between refractory harzburgites and supercritical C-O-H fluids equilibrated with sediments should be performed. *We did show the results of this mixing calculation in Figure 9b (previously 7b) but have replaced it with a comparison of the predicted metasomatic agent in equilibrium with garnets in the orthopyroxene-rich harzburgites. This shows that they are remarkably similar to mixing of refractory harzburgite with supercritical fluids equilibrated with sediments (based on the study of Sokol et al., 2023 & 2024).*
- 4) Generally, F compatibility in mantle silicate minerals follows the order: garnet > pyroxene > olivine. Interestingly, the Kaapvaal Craton Opx-rich harzburgites show the opposite trend, with olivine exhibiting much higher F contents than pyroxenes. The authors need offer an explanation for this observation. *There are very few data on F in mantle minerals and we are one of the first groups to generate this data for cratonic mantle. Our work shows that the main host of F is olivine followed by the pyroxenes (see lines 147-159).*
- 5) Fluids are generally enriched in Cl relative to F, as also suggested by Fig. 57 in the manuscript. Is it possible to reconstruct the bulk Cl contents to further investigate the role

of supercritical C-O-H fluids? Chlorine is not compatible in mantle olivine and the pyroxenes and so it is not possible to reconstruct bulk Cl compositions for most of the peridotite xenoliths in our study. Chlorine does, however, partition into phlogopite and so we cite the Cl data where this phase is present (Lines 164 & 331).

6) I also noticed some discussion about alkalinity addition through diamondiferous orthopyroxene-rich garnet harzburgites, but more details are more helpful. Evidence of K addition is present from the phlogopite in the diamondiferous orthopyroxene-rich garnet harzburgite. The phlogopite also has relatively high Na and Cl contents as mentioned in the text (Line 164). I agree this is important and have added the following sentence in Line 337 of the main text: 'The presence of phlogopite with relatively high Na contents in the most orthopyroxene-rich garnet harzburgites is evidence of the alkaline nature of the percolating silicic fluids.'

7) The methods section provides detailed information on volatile measurements using SIMS, but it does not address the reliability of the results. Some standard references as blind samples should be used to estimate the reliability of their results. Unfortunately, there are no sets of SIMS standards for olivine. At the time of analysis the most appropriate standard was ALV-519-4-1 (<https://georem.mpch-mainz.gwdg.de>). This is a basaltic glass from the FAMOUS area of the Mid Atlantic Ridge (Langmuir et al., 1977). For H₂O backgrounds, we also analysed San Carlos olivine (see Aubaud et al., 2015) and OL-116610-5. Analyses of these standards have been added to Supplementary Table 3. All of the data presented in our paper were run at the Edinburgh Ion Microprobe Facility (EIMF) using the same method and so internally consistent. Mantle olivines and pyroxenes separated from some of the same Kaapvaal peridotite xenoliths have been analysed by SIMS at EIMF and also by FTIR at Bayreuth (using the Withers 2012 calibration; Heckel, 2023). The table below shows the data are comparable. This has now forms Table S2 in the Supplementary material.

	Olivine		Orthopyroxene	
	This work	Heckel (2023) ³⁰	This work	Heckel (2023) ³⁰
	SIMS (Edinburgh)	FTIR (Bayreuth)	SIMS (Edinburgh)	FTIR (Bayreuth)
BD2128		22 30	191	155
BD2170		44	183	180

I hope these suggestions help improve the

manuscript. Minor comments

Line 823. elements to element. This typo has been corrected

Line 853.% to wt.% This has been corrected to volume %

Line 872. What are the residues predicts? Melting residues This typo has been corrected

Line 888. MORB source normalized? This typo has been corrected

Line 913. is from Ref. 100. 'Ref. 100' has been added

Reviewer #2 (Remarks to the Author):

The role of C-O-H-F-Cl fluids in the making of Earth's continental roots

This manuscript by Gibson et al. addresses a longstanding and significant open question in mantle geochemistry regarding the origin of Si enrichment observed in some of Earth's cratonic 'roots.' Since Boyd's initial observations in 1989, this phenomenon has been the focus of numerous

studies, leading to two primary schools of thought: 1. orthopyroxene enrichment is attributed to be a key characteristic inherited from a nonpyrolytic mantle, or 2. Secondly, silicon enrichment in the form of excess orthopyroxene (opx) occurs in preexisting olivine-rich melting residues, such as metasomatism in subduction zones or plume settings. Overall, the entire manuscript is a pleasure to read and very clear.

The authors tackle this question by analyzing a suite of mantle xenoliths and their minerals for volatile contents, including H₂O, Cl, and F, with a particular focus on the opx-rich xenoliths from the Kaapvaal Craton, including one diamond-bearing opx-rich harzburgite. The authors argue that

based on the observed volatile enrichment of olivines within the opx-rich xenoliths, the most likely cause of the opx enrichment was the infiltration of a small volume of supercritical C-O-H-F-Cl-bearing fluids sourced from subducted oceanic lithosphere ± sediment and its reaction with a previously depleted mantle to form the observed Si enrichment. The above is in contrast to recent thermodynamic studies, which propose a reaction between "dry" komatiitic melts and a melt-depleted lithospheric mantle at high pressure, forming opx in a peritectic reaction (e.g., Tomlinson and Kamber 2021).

The Cambridge group is one of the pioneering teams in the field and is an expert in the analysis of volatiles in mantle minerals and their storage at lithospheric conditions. This large body of work on the volatile concentrations (F, H₂O, and Cl) on 22 xenoliths and 65 mineral separates coupled with petrographic major and trace element analysis represents a very impressive analytical campaign and achievement. In my opinion, this contribution is vital to the community and keeps the long-lasting debate alive.

After some moderate to significant revisions, this study should be published in Nature Communication

s. Major remark

- 1) It appears that the entire model is heavily reliant on a single diamond-bearing, opx-rich harzburgite with approximately 120 ppm F, along with two additional xenoliths, all from the Bultfontein locality. While the scarcity of suitable samples is understandable, drawing robust correlations from only three data points is challenging. This is not strictly true. The manuscript focuses on the volatile

contents of a suite of mantle peridotites from the Kaapvaal craton. The diamond-bearing peridotite (BD2125) represents an end-member in the xenoliths that have excess orthopyroxene. We believe that the rarity of diamond bearing peridotites (there are probably <10 that have been reported in the published literature) makes this sample especially important and our findings novel.

- 2) Additionally, making a global assessment based on such limited data, which primarily pertains to the Kaapvaal Craton, seems to be overreaching. While our work focuses on the origin of excess orthopyroxene in the Kaapvaal (and to a lesser extent the Tanzania craton) it has long been known that such enrichments are present in numerous other global cratons. Although our multiple volatile dataset is unique in addressing the cause of the excess orthopyroxene, we hope that our work will now drive forward similar analyses in xenolith suites from other cratons.
- 3) Given the long residence time of these xenoliths in the mantle, a more detailed explanation is needed to clarify how the samples retained such high F contents over geological timescales. This is beyond the scope of the current manuscript but something we aim to address in a long format journal. In essence, the diffusion of F in olivine is very slow and similar to Ti.

Specific remarks in the text

Abstract

The abstract is well written very lays out the question very elegantly, and the novelty of the study. Line 13: Why mention Siberia and Rae and Tasmania cratons here? It is a bit misleading in my mind. These cratons are mentioned to highlight that the enrichment of the lithospheric mantle in orthopyroxene is not restricted to the Kaapvaal craton, i.e. this is a global phenomenon. We have now emphasized the global occurrence of this mantle lithology in the Discussion by adding:

'While our study has primarily focused on the association of volatiles and orthopyroxene-enrichment in the Kaapvaal craton mantle, where some of the greatest excess orthopyroxene and silica has been observed, similar mantle peridotites have been widely reported from the Siberian, Slave, Rae and Tanzanian cratons ^{e.g. 5,9-17}. There are, however, no F or Cl data for olivine and pyroxenes in well-characterised peridotites from these other cratons and systematic analytical work is now required to firmly establish if the enrichment in multiple volatiles and orthopyroxene is a global phenomenon.'

Line 14: What is the significance of the Ti depletion? Maybe highlight this observation a bit more here? Yes, this is a good point. I have modified this sentence to read 'This suggests infiltration by super-critical C-O-H fluids -- rich in silica, fluorine and chlorine and depleted in Ti -- fluxed from subducted oceanic lithosphere (carbonated pelites, eclogites and serpentinites) was important in driving olivine-to-orthopyroxene transformation and diamond formation under reduced conditions during the Archean.'

Line 20: A bit vague, in my opinion. Agreed. This sentence has been changed to read 'To further investigate the role of melts and fluids, we analysed volatile (H₂O, F, Cl) contents in Kaapvaal craton peridotites'

Introduction

Line 48: Mg# notation is explained in line 50. I would move it here. Agreed. This definition has been moved.

Results

Major and trace element chemistry

Line 110: What is the significance of phlogiopite? Is this phase related to kimberlite metasomatism, or is it more ancient? There is no evidence of kimberlite-related metasomatism in this sample and we believe that the phlogopite formed at the same time as the excess orthopyroxene (see experiments of Gao et al. (2021)).

Volatile elements

It would be helpful to visualize the different populations and how they relate to one another, perhaps using a box-and-whisker diagram similar to the one in the supplemental file. Additionally, many of the measurement error bars appear to overlap. If this is typical in the field of volatiles in the mantle, an explanation of why the results remain robust would be beneficial. For example, in lines 144-146, there is a clear population difference, and this should be represented in a figure to strengthen the argument. Thank you for this suggestion. A new set of box and whisker plots has been included in the main text (new Figure 6) to show the variations in bulk volatile contents of different Kaapvaal lithologies. There is also a new set of box and whisker plots in the Supplementary material (Figure S6) to show variations of volatiles in olivine and pyroxenes. Both help visualize the variations and show how the standard deviations vary between lithologies. The 'boxes' show where 75% of the data fall and that there are systematic variations between the reconstructed volatile contents of the different mantle peridotite lithologies.

Bulk volatile content

Line 185: How robust are the correlations that are not shown? This seems like a critical point to address. Additionally, are there correlations with other minor or trace elements in olivine and orthopyroxene? It might be interesting to examine first-row transition elements such as Zn, Sc, V, and Mn. Because of the relatively small number of samples it is not possible to undertake a rigorous statistical analysis of bulk compositions, so these are qualitative observations. The correlations of volatiles with minor elements in olivine and orthopyroxene (where the dataset is larger) is beyond the scope of the current study and something that we are actively working on.

The current projection in Figure 5, illustrating the relationship between bulk LREE versus F and the Sm/Nd peak position, could be improved, as it appears somewhat misleading. Alternative projections, such as plotting :LLREE (ppm) versus F (ppm), Si, Mg, or Nd/La versus F, Si, Mg, might provide clearer insights. It might also be worth considering a more integrated presentation of panels A and B to better highlight the differences between the opx-rich and

opx-poor samples.

I think the reviewer is referring to Figure 3 rather than Figure 5 in the original version of the manuscript? This plot was chosen to show the variability in bulk rare-earth element abundances. We have removed the text 'Si, Mg, C & H+F' from the upper plot as we agree this is misleading.

Mantle Melting Models

I think this and subsequent sections are part of the discussion, but it is not noted in the text. Agreed. 'Discussion' has now been added as a sub-heading to make this clearer.

Line 193: I believe you meant 2b and not 2a? Thank you – this typo has been corrected

Additionally, I found the use of the color gradient a bit excessive. If there is a strong correlation, it might be worth dedicating its own panel-just a suggestion.

The color gradient is used in Figure 2b to highlight: (i) high Mg samples have the highest contents of Bulk H₂O + F; and (ii) how this differs from xenoliths that follow the Phanerozoic melt extraction trend. We would like to keep the figure and colourbar as it is.

Line 192-193: Fo. Molar proportions, yes? Yes, thank you for spotting this typo.

Reactive infiltration of Al-enriched komatiites

My main comment here is that the proposed fluid-associated opx addition is comparable to melt addition, except for one sample that is a bit strange and needs to be addressed. The Ti and HREE mismatch is very interesting. I have re-modelled the trace element data so that it now focuses on metasomatic agents in equilibrium with garnet in the diamondiferous orthopyroxene-rich harzburgite. This is a more appropriate approach but does not permit us to model the volatiles because of a lack of partitioning data and analyses for garnet.

Infiltration of subduction-related melts and fluids

Line 249-252 I think a citation is needed for this statement. Agreed, I've added examples of relevant publications, i.e. Kessel (2005), Meltzer & Kessel (2023) and Chen et al. (2023).

Why did you choose the fluid composition of Sokol et al.? -1100°C at -8 GPa seems a bit cold for Archean tectonics. What geotherm does this plot on? It might be worth looking at the Timmerman et al. (2022) EPSL paper on Mesoarchean diamonds to see if their thermal modeling could explain the cold and transient conditions relevant to your proposed fluids. There are very few experiments that have looked at the trace and volatile (H, F & Cl) contents of supercritical melts and fluids in subduction zones. The results of Sokol et al. (2024) are appropriate for the temperature and pressure in the down-going slab rather than the cool conductive geotherm of the Kaapvaal cratonic root. Sokol's experiments were conducted over 2 GPa intervals but we chose the high pressure (7.8 GPa) and temperature (>940 °C) results because monazite and phengite were absent in the run products, i.e. had broken down and contributed to the trace element budgets of the super-critical fluids.

Is there additional evidence supporting monazite dissolution? The explanation for subaerial weathering seems a bit too convenient. This section has been removed from the revised text.

It might also be valuable to explore natural examples of silicic fluids trapped in diamonds as potential candidates for the required supercritical fluid. The recent compilation by the HUJI group on fluid compositions in diamonds could be relevant. While there is no F content data for these inclusions due to hydrofluoric acid use during sample prep, H₂O, Cl, Si, Al, and other trace elements are well-characterized and are known metasomatic agents. This is a helpful suggestion, thank you. I have looked into the available data and undertaken some new modelling related to the compositions of the diamond inclusion fluids published by the HUJI group (see Figure 9c).

Regarding diamond-forming fluids, to my knowledge, there are no fibrous diamonds from the Kaapvaal of this age that are SiO₂-rich. However, their absence in the diamond record does not necessarily exclude their involvement. The oldest observed fluids in the Kaapvaal are carbonatitic, which cannot account for Si enrichment. You might consider looking at Weiss et al. (2021, Nature Communications) on how fibrous diamonds track the metasomatic history of the Kaapvaal using helium isotope systematics. This is a very interesting paper, which I have re-read, and note that the oldest observed fluids reported by Weiss et al. (2021) are younger (Proterozoic) than those that we interpret from the petrology and geochemistry of the orthopyroxene-rich garnet harzburgites. As the reviewer indicates, a lack of data does not preclude the presence of SiO₂ rich fluids in the Archean.

The "smoking gun," in my view, would be high-Mg olivine in diamonds with high F, Cl, and H₂O. While this might be beyond the scope of this study, it is worth considering.

Yes, agreed. There are some olivines in diamonds with high F contents but the data is presented in an unpublished PhD thesis and Edinburgh ion probe report.

On that note, olivine inclusions from the De-Beers pool kimberlites have a distinct high-Mg number and could provide valuable insights (see Stachel et al., 2022, RIMG review, Figure 6). Yes, this is interesting, and we have now included a sentence related to the De-Beers pool diamond inclusions in the 'Discussion'.

I'm not sure that the segway to the next section is warranted. Maybe lines 279-300 should be incorporated here, and sections Infiltration of subduction-related melts and fluids & Carbonation, hydration, fluorination, and chlorination of cratonic mantle should be renamed and restructured. This now forms a sub-section in the 'Discussion' section.

Line 289: The major obstacle is not just oxygen isotopes but also mass balance considerations and the cold geotherm that you just noted above. We now show that the mass balance considerations are not a problem with the revised trace-element modelling.

Line 329: The Slave and Kaapvaal cratons have very different geological histories. Before reaching this conclusion, Slave xenoliths should be added. Agreed. I have replaced this sentence with 'Further support for the idea of recycling of sediment in the Archean beneath the Kaapvaal craton is provided by sulfur isotope ratios of sulfide inclusions in diamond'.

Line 344: Why is this seen in Fig S3? Maybe you mean S4? Thank you – this typo has been corrected

If so, I think this (or a version of it) should be moved to the main text. The original Fig. S3 has been moved to the main text.

Line 344-347: This claim warrants a figure in my mind. I think this is clear without a figure, given the journal restrictions and that we have already added several new figures.

Also, it is worth noting that the K and (F) in K-rich phases are likely lost during subsequent melting. We do not believe that there has been any subsequent melting of the mantle preserved in these xenoliths.

Lines 364-376: The link to wider implications is very interesting but a bit of an overpromise and underdeliver, especially for the North China Craton. Is there any insight as to why the opx enrichment is restricted to the Kaapvaal Slave and Siberian Cratons?

It is also apparent in the Tanzania craton. It may be that it is even more widespread but has not been observed in mantle xenolith suites from other, less well-studied cratons.

Some other remarks

The experimental results of Gao et al. (2021) might be worth considering aside from the introduction as they are similar to what the authors propose. The experiments of Gao et al. (2021) are interesting but were undertaken at low pressures (1.5 GPa) and inconsistent with the presence of diamond in our sample suite. On the basis of these, Gao et al. (2021) suggest that orthopyroxene enrichment in the Kaapvaal craton mantle may be explained by a two-stage 'metasomatism-depletion' model. The second melting event is required to generate the high Mg# and quite different to our model. The presence of phlogopite in experimental run products generated by Gao et al. (2021) at <1100 °C and its absence at >1200 °C is, however, consistent with previous experiments on phlogopite stability and the temperature threshold that we invoke in our model. A reference to this has now been included in the main text.

What about using the delta Mg/Si notation of Canil and Lee 2009? We preferred to use Bulk MgO/SiO₂ rather than the delta Mg/Si notation.

You did something similar in supp Fig. S4 but did not add volatiles to the color bar. It seems from Fig. 2 that there are no robust correlations. The new box and whisker plot (Figure 6) shows that there are correlations between lithological type and reconstructed bulk H + F content. This supports the information on Figure 3b.

Don't you expect that the more the xenolith departs from the Boyd trend, the more fluid reacts, and thus, the more F in the olivine? Here, some diffusion modeling might come in handy. This would be an excellent calculation to undertake, and something that we have considered previously but unfortunately there are no F diffusion data for olivine.

Why does BD1153 (dunite) have so much F in it? Fluorine strongly partitions into olivine and this is a metasomatised dunite.

Figures and supplementary materials remarks

The Figures overall are very engaging. I have a couple of remarks and possible suggestions.

Fig. 1: Since the focus is primarily on the Kaapvaal Craton, could you include a panel showing the location of the retrieved xenoliths? Or provide it in the supplement? Is there any observed spatial variation? This is a good idea. A map showing the locations of samples in the Kaapvaal craton has been included in the Supplementary material (Fig. S3). There is no spatial variation in volatile contents.

Fig. 2 Panel A might be redundant; I think that Panel B is the star of this figure. We included Panel A because this shows the original concept proposed by Boyd (1989), which the broader readership of Nature may not be familiar with.

Fig. 3 See comment above. Plenty of projections might be better utilized to illustrate the differences you are discussing. Please see our response above.

Why normalize to MORB source and not Primitive mantle? From the way I read the manuscript, it is not clear why you chose to show this normalization. This normalization is used because cratonic mantle is widely believed to represent residues of large amounts of melting. We believe that this is more appropriate to model subsequent depletion and enrichment events than primitive mantle. This is now explained in the caption for Figure 4 as follows 'A MORB-source normalization is used because cratonic mantle is widely believed to represent residues of large amounts of melting. MORB source normalisation factors are from Salters & Stracke ³⁴.'

Fig. 5. Are there any inclusions in the diamond to plot on this diagram? Not to our knowledge

Why don't you show the olivine from the opx-rich sample in panel a? It is a bit misleading. If the two plots are combined the figure becomes overwhelming and the impact of the various trends is lost, hence our decision to have two figures.

In panel b, the diamond-bearing opx-rich sample BD2125 has 120 ppm F in the supplementary table and is plotted as -138 ppm in the figure. Am I missing something? Is there any meaning to the correlations? Sincere apologies. The incorrect table was uploaded to the Supplementary Material and has now been corrected.

Figure 8 is great, but it has been recycled from other publications. If you want to show a conceptual model, I believe it merits a new figure. This diagram fully supports our findings and we would like to keep it as it is.

The supplemental tables need a bit more work to make them clear to the reader. missing descriptions. In the supplementary table of the bulk trace element compositions, there is no table description indicating which samples are from the Kaapvaal, Tanzania, or off-craton. For those who are not familiar with the local names, it is hard to follow. Agreed. This information has now been added.

Also, can you group them in a systematic order? They are grouped according to location. This wasn't apparent before but should be now (please see response to comment above).

Supp table Bulk trace element compositions There is data in columns O and P that should be deleted. Thank you. This typo has been corrected.

The clinopyroxene table needs a description. Agreed. A caption to this dataset has now been added.

I have no remarks on the Methods section

Best,
Oded Elazar

Reviewer #3 (Remarks to the Author):

Review of the manuscript "The role of C-O-H-F-Cl fluids in the making of Earth's continental roots" by Gibson, Jackson, Crosby, and Day, submitted to Nature Communications

The focus of this manuscript is the existence of opx-rich garnet harzburgites in the SCML. The authors examine 3 scenarios for the formation of excess opx in these xenoliths: high-pressure melting residues, komatiite interactions, or subduction-related silicic melts and fluids. The authors analyze olivine, pyroxene and garnet in different xenoliths for H₂O, F and Cl content and conclude that the best model is the interaction of the harzburgitic mantle with C-H₂O-F-Cl fluids derived from dehydration of the subducting slab.

The formation of opx-rich harzburgites is a long-standing question. The suggestion that these xenoliths were formed due to the interaction of depleted harzburgite with slab-derived silica-rich fluids or melts is not new. However, this manuscript attempts to provide additional support for this model via the analyses of volatiles in the peridotite minerals themselves, and provides unique information for the type of fluid/melt interacting with the harzburgite (volatile-rich). This manuscript is important and very interesting.

The manuscript is well written and easy to follow.

Major comments and thought

1. Opx-rich harzburgites are mainly found in Slave, Rae, Kaapvaal and Siberia cratons. Does that mean that fluids interacted with the SCML only in these regions and no other SCML? This is an interesting point. Mantle xenoliths are localized in their distribution and it is interesting that the orthopyroxene-rich ones are found in cratons where the largest numbers of mantle xenoliths have been found and analysed to date. It doesn't, however, negate orthopyroxene-rich mantle being present elsewhere. They may well just not have been found.
2. There is no discussion on carbon in these potential fluids. More information on C in these fluids has been added to the Discussion section.
3. Saline HDF (found in fluid inclusions in diamonds) were identified in many cratons around the world (see review by Weiss et al. 2022). Not just were there is excess opx. Also, Weiss and others suggested that silicic HDFs originate from the dehydration of the subducting slab. It will be interesting to use the trace element compositions of either the saline HDF or silicic HDF instead of experimental supercritical fluid in Figure 7. Thank you, this is a helpful suggestion and we have now explored these datasets. Silicic and saline HDFs have variable concentrations of trace elements. There are no data for volatiles but variations in normalized [Zr/Hf] and [Sr/Nd] suggest that the fluids associated with the orthopyroxene-rich harzburgites may be comparable to silicic low Mg carbonatitic HDFs. Some text and a figure illustrating this is now included in the Supplementary Material.
4. Elazar and Kessel (2024) interacted harzburgite with eclogite in a hydrous environment (CHO fluid) at the conditions above the eclogitic wet solidus, but below the wet peridotite solidus. At the experimental conditions, eclogitic supercritical fluid was interacting with the harzburgite. During

this interaction, the amount of ol decreases while the amount of opx increases. Their work may apply to the suggested model. Agreed, this work is relevant to our study and we have cited it at various places in the manuscript. Unfortunately, only a few trace elements were analysed and so it is not possible to use this data in our numerical models. We have also looked at the data for eclogitic fluids in Tsay et al. (2017) and Rustioni et al. (2021) but they have positive anomalies at Sr and so inconsistent with our observations.

Minor comments

5. Line 90- The authors use Kessel et al. (2005) (ref. 35) for trace elements profiles during enrichments. Kessel et al. (2005) focuses on trace elements in eclogite+H₂O system. It is better to refer to Kessel et al. (2015): The behaviour of incompatible elements during hydrous melting of metasomatized peridotite at 4-6 GPa and 1000 DC-1200 oc. Thank you. This citation has been changed

6. Line 116 – REE profiles are in Figure S5 (not S4). This typo has been changed to S5.

7. Lines 131- Please check the calculated averages for the H₂O and F in the various minerals and in the bulk peridotites. Following the data presented in Table 51, I calculate different values. For example, The H₂O content in olivine in opx-rich garnet harzburgite in Table 51 are 37,69 and 69 ppm. The average is 58±18 ppm (and not 49.5±68 as states in the text). Same for opx etc. Mea culpa! I uploaded the incorrect table to the Supplementary Material. The discrepancy arose because Table S1 shows our new data whereas the averages stated in the text included published analyses for the Kaapvaal. Thank you for noticing this typo, which has now been corrected. I have also included two additional spreadsheets (Supplementary Tables 5 & 6) that show the means, standard deviations, median values and number of counts of the mineral volatile contents and bulk xenolith volatile contents cited in the manuscript.

8. Line 191-193 – Figure 2a does not show bulk H₂O+F content. Figures 2b does. The authors declare that the Phanerozoic oceanic melt residues show a decrease in bulk H₂O+F with increasing Fo content and model abundance of ol. Thank you. This has been changed.

9. Line 236-237 – Ti seems to fit well with the infiltration of Al-enriched komatiites model. Ba and Rb do not fit. This has been changed.

10. Figure 7 – In the legend, the reference is Sokol et al (2024). In the figure caption, it is Sokol et al. (2023). Thank you. This has been changed.

11. The fluid taken from this reference is a supercritical fluid in equilibrium with a pelite. Please explain this choice. Why not choose a supercritical fluid in equilibrium with eclogite

Unfortunately, we have not been able to find appropriate trace element and volatile data for a supercritical fluid in equilibrium with eclogite. This is despite an extensive

literature search and reaching out to others in this field.

12. Check the symbol of the supercritical fluid from Sokol in Fig. 7b. (Triangle in the legend, circles in the figure). Thank you. This symbol has been changed.

AUTHORS RESPONSES TO REVIEWERS' COMMENTS

In green Authors replies to comments in comments in black (Reviewers 1 & 2) or red (Reviewer #3)

Reviewer #1 (Remarks to the Author)

Thank you to the authors for addressing my concerns. I am happy to recommend this version for publication, subject to the following minor suggestions:

I suggest changing "continental roots" in the title to "cratonic roots" for greater precision.

We would like to keep the original title

Line 23: Please specify the type of volatiles being referred to.

This information has been added

Line 135: Please note the lowest Ce/Pb ratios observed.

This information has been added

Reviewer #2 (Remarks to the Author)

The authors have addressed all of my previous remarks, and the revised manuscript looks excellent!

Just one thought that came to mind during my second read (and apologies if I missed this)

Do you have any nitrogen aggregation data on the diamonds hosted in the diamondiferous orthopyroxene-rich garnet harzburgite (BD2125)?

Although previous work has clearly shown that these diamonds are old, it would be a nice addition if you have nitrogen aggregation systematics—especially if the diamonds have high nitrogen contents and show significant aggregation. This could further strengthen the interpretation of an ancient subduction-related origin.

I'm afraid that there is no nitrogen data available for the diamond hosted in BD2125. It would be fabulous if this type of study could be undertaken on other diamonds found in orthopyroxene-rich harzburgites.

I also took a moment to look for any remaining typos, and everything looks good to me. I trust the editorial team will give it one final check before publication.

Other than that, the paper looks great!

All the best,
Oded Elazar

Reviewer #3 (Remarks to the Author)

I accept all the changes and replies made by the authors, except a few small corrections, as detailed below.

In black – the original comment
In Blue – the authors reply
In Red – my reply to their reply

4. Elazar and Kessel (2024) interacted harzburgite with eclogite in a hydrous environment (CHO fluid) at the conditions above the eclogitic wet solidus, but below the wet peridotite solidus. At the experimental conditions, eclogitic supercritical fluid was interacting with the harzburgite. During this interaction, the amount of ol decreases while the amount of opx increases. Their work may apply to the suggested model.

Agreed, this work is relevant to our study and we have cited it at various places in the manuscript. Unfortunately, only a few trace elements were analysed and so it is not possible to use this data in our numerical models. We have also looked at the data for eclogitic fluids in Tsay et al. (2017) and Rustioni et al. (2021) but they have positive anomalies at Sr and so inconsistent with our observations.

Although the authors agreed as to the relevance of Elazar and Kessel (2024) work and wrote that it is cited and discussed in the manuscript, I do not see it is cited. Instead, the study of Elazar et al. (2019) is cited. This is also a relevant study on the formation of silicic HDFs, however, it focuses on the interaction of H₂O-CO₂ fluids with eclogites while the study of Elazar and Kessel (2024) discusses the enrichment of OPX due to the interaction of harzburgite with eclogite in a hydrous environment. I believe Elazar and Kessel (2024) is more relevant to the model suggested in this study.

Thank you, this reference has been changed to Elazar & Kessel (2024)

5. Line 90- The authors use Kessel et al. (2005) (ref. 35) for trace elements profiles during enrichments. Kessel et al. (2005) focuses on trace elements in eclogite+H₂O system. It is better to refer to Kessel et al. (2015): The behaviour of incompatible elements during hydrous melting of metasomatized peridotite at 4-6 GPa and 1000 DC-1200 oc.

Thank you. This citation has been changed

The authors cited the wrong Kessel et al. (2015) paper. They cited the work titled [Melting of metasomatized peridotite at 4–6 GPa and up to 1200 °C: an experimental approach. *Contrib Mineral Petrol* **169**, 37 (2015)] which does not deal with trace elements. Instead, the relevant Kessel et al. (2015) study is titled [The behaviour of incompatible elements during hydrous melting of metasomatized peridotite at 4-6 GPa and 1000 DC-1200°C. *Lithos*, 236-237, 141-155]. This study focuses on trace elements, relevant to the current manuscript.

Thank you, this reference has been corrected

11. The fluid taken from this reference is a supercritical fluid in equilibrium with a pelite. Please explain this choice. Why not choose a supercritical fluid in equilibrium with eclogite /

Unfortunately, we have not been able to find appropriate trace element and volatile

data for a supercritical fluid in equilibrium with eclogite. This is despite an extensive literature search and reaching out to others in this field.

The changes made to Figure 9 (originally 7) as well as the relevant text are a welcome improvement. The only small comment is that in the revised caption there is now an unclear sentence (a part of it is missing? which was provided in the original caption of this figure):

Thank you, the figure caption has been revised to remove the typo.

Sally A Gibson
22nd July 2025

Comments to comments:

I accept all the changes and replies made by the authors, except a few small corrections, as detailed below.

In black – the original comment

In Blue – the authors reply

In Red – my reply to their reply

4. Elazar and Kessel (2024) interacted harzburgite with eclogite in a hydrous environment (CHO fluid) at the conditions above the eclogitic wet solidus, but below the wet peridotite solidus. At the experimental conditions, eclogitic supercritical fluid was interacting with the harzburgite. During this interaction, the amount of ol decreases while the amount of opx increases. Their work may apply to the suggested model.

Agreed, this work is relevant to our study and we have cited it at various places in the manuscript. Unfortunately, only a few trace elements were analysed and so it is not possible to use this data in our numerical models. We have also looked at the data for eclogitic fluids in Tsay et al. (2017) and Rustioni et al. (2021) but they have positive anomalies at Sr and so inconsistent with our observations.

Although the authors agreed as to the relevance of Elazar and Kessel (2024) work and wrote that it is cited and discussed in the manuscript, I do not see it is cited. Instead, the study of Elazar et al. (2019) is cited. This is also a relevant study on the formation of silicic HDFs, however, it focuses on the interaction of H₂O-CO₂ fluids with eclogites while the study of Elazar and Kessel (2024) discusses the enrichment of OPX due to the interaction of harzburgite with eclogite in a hydrous environment. I believe Elazar and Kessel (2024) is more relevant to the model suggested in this study.

5. Line 90- The authors use Kessel et al. (2005) (ref. 35) for trace elements profiles during enrichments. Kessel et al. (2005) focuses on trace elements in eclogite+H₂O system. It is better to refer to Kessel et al. (2015): The behaviour of incompatible elements during hydrous melting of metasomatized peridotite at 4-6 GPa and 1000 DC-1200 oc.

Thank you. This citation has been changed

The authors cited the wrong Kessel et al. (2015) paper. They cited the work titled [Melting of metasomatized peridotite at 4–6 GPa and up to 1200 °C: an experimental approach. *Contrib Mineral Petrol* **169**, 37 (2015)] which does not deal with trace elements. Instead, the relevant Kessel et al. (2015) study is titled [The behaviour of

incompatible elements during hydrous melting of metasomatized peridotite at 4-6 GPa and 1000 DC-1200°C. *Lithos*, 236-237, 141-155]. This study focuses on trace elements, relevant to the current manuscript.

11. The fluid taken from this reference is a supercritical fluid in equilibrium with a pelite. Please explain this choice. Why not choose a supercritical fluid in equilibrium with eclogite /

Unfortunately, we have not been able to find appropriate trace element and volatile data for a supercritical fluid in equilibrium with eclogite. This is despite an extensive literature search and reaching out to others in this field.

The changes made to Figure 9 (originally 7) as well as the relevant text are a welcome improvement. The only small comment is that in the revised caption there is now an unclear sentence (a part of it is missing? which was provided in the original caption of this figure):